# E-cadherin integrates mechanotransduction and EGFR signaling to control junctional tissue polarization and tight junction positioning

Matthias Rübsam[1,2,3], Aaron F. Mertz[4,11], Akiharu Kubo[5], Susanna Marg[6], Christian Jüngst [2], Gladiola Goranci-Buzhala[1,2,3], Astrid C. Schauss[2], Valerie Horsley[7], Eric R. Dufresne[4,8], Markus Moser[9], Wolfgang Ziegler [6], Masayuki Amagai [5], Sara A. Wickström[2,10] & Carien M. Niessen[1,2,3]

Generation of a barrier in multi-layered epithelia like the epidermis requires restricted positioning of functional tight junctions (TJ) to the most suprabasal viable layer. This positioning necessitates tissue-level polarization of junctions and the cytoskeleton through unknown mechanisms. Using quantitative whole-mount imaging, genetic ablation, and traction force microscopy and atomic force microscopy, we find that ubiquitously localized E-cadherin coordinates tissue polarization of tension-bearing adherens junction (AJ) and F-actin organization to allow formation of an apical TJ network only in the uppermost viable layer. Molecularly, E-cadherin localizes and tunes EGFR activity and junctional tension to inhibit premature TJ complex formation in lower layers while promoting increased tension and TJ stability in the granular layer 2. In conclusion, our data identify an E-cadherin-dependent mechanical circuit that integrates adhesion, contractile forces and biochemical signaling to drive the polarized organization of junctional tension necessary to build an in vivo epithelial barrier.

[1] Department of Dermatology, University of Cologne, Cologne 50931, Germany. [2] Cologne Excellence Cluster for Stress Responses in Ageing-associated diseases (CECAD), Cologne 50931, Germany. [3] Center for Molecular Medicine Cologne (CMMC) University of Cologne, Cologne 50931, Germany. [4] Department of Physics, Yale University, New Haven, CT 06520, USA. [5] Department of Dermatology, Keio University School of Medicine, Tokyo 160-8582, Japan. [6] Hannover Medical School, 30625 Hannover, Germany. [7] Department of Molecular, Cellular, and Developmental Biology, Yale University, New Haven, CT 06520, USA. [8] Departments of Mechanical Engineering and Materials Science, Chemical and Environmental Engineering, and Cell Biology, Yale University, New Haven, CT 06520, USA. [9] Max Planck Institute for Biochemistry, Am Klopferspitz 18, Martinsried 82152, Germany. [10] Paul Gerson Unna Group 'Skin Homeostasis and Ageing', Max Planck Institute for Biology of Ageing, Cologne 50931, Germany. [11] Present address: Laboratory of Mammalian Cell Biology and Development, The Rockefeller University, New York, NY 10065, USA. Correspondence and requests for materials should be addressed to C.M.N. (email: carien.niessen@uni-koeln.de)

The ability to build asymmetry in molecular, cellular, and tissue structures is a fundamental property of development, growth, and regeneration and is essential for tissue function. For example, apico–basolateral polarity promotes the asymmetric positioning of intercellular junctions and the cytoskeleton within the cell to drive functional barrier formation in simple epithelia[1]. In contrast, virtually nothing is known about how multi-layered epithelia such as the skin epidermis or the esophageal epithelium establish polarized barriers. A fundamental hallmark of the epidermis is the confinement of barrier-forming TJs only to the most apical viable layer (Fig. 1a), indicating that basal-to-apical polarization occurs at the tissue level. However,

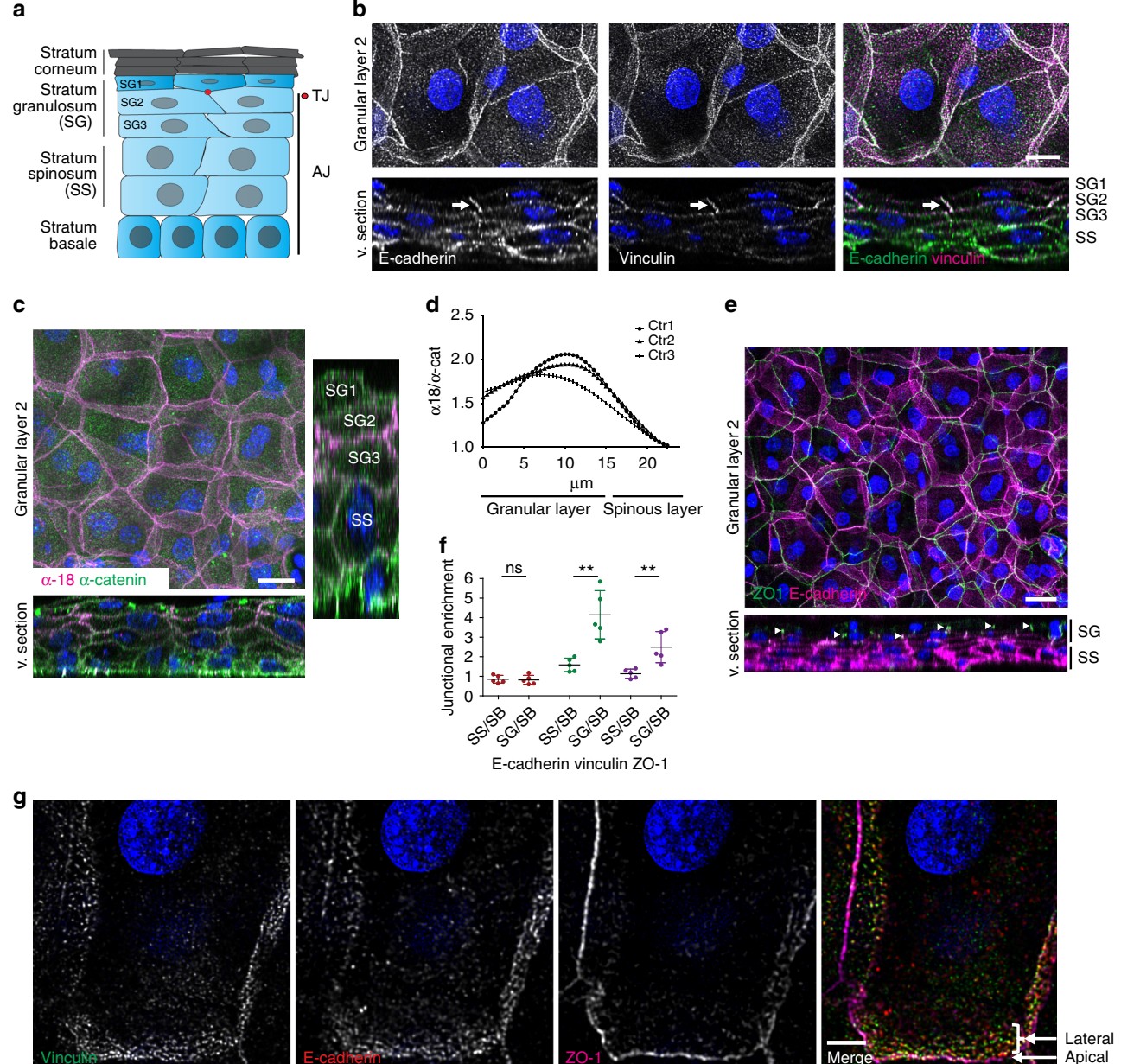

**Fig. 1** Intercellular junctions are polarized across and within epidermal layers. **a** Cartoon depicting epidermal structure and distribution of junctions in different epidermal layers (TJ tight junctions; AJ adherens junctions). **b** Newborn epidermal whole-mount immunofluorescence analysis for tension-high (vinculin) adherens junctions (E-cadherin). Note that vinculin-positive junctions are primarily present in the granular layer 2 (SG2). Scale bar, 10 μm. **c** Newborn epidermal whole-mount immunofluorescence analysis for total and tension-sensitive epitope (α18) of α-catenin showing that mechanosensitive α-catenin is only present in SG2. The right panel is a cross-sectional view stretched in Z direction to better visualize single layers. Scale bar, 20 μm. **d** Quantification of fluorescence intensity ratios in **c** of α18 vs. total α-catenin in different layers. Curves show mean ratios from three stack profiles per biological replicate (*n* = 3). **e** Staining for E-cadherin (AJ) and ZO-1 (TJ) (arrowheads) showing TJ enrichment in SG2. **f** Quantification of whole-mount junctional intensities of E-cadherin, ZO-1, and vinculin in different epidermal layers. Values were normalized to basal layer intensity. Mean values of 5 biological replicates are shown (dots, 10 junctions per replicate), *P* = 0.8105 (E-cadherin); 0.0064 (ZO-1); 0.0021 (Vinculin) with Student's *t*-test, *n* = 5. Scale bar, 20 μm. **g** SG2 projections from apical to subapical cell–cell junctions stained for vinculin, E-cadherin, and ZO-1 showing an basolateral AJs network and an apical TJ network. Scale bar, 5 μm. **b**, **c**, **e**, **g** Deconvolved confocal stack projections and orthogonal views (virtual section; v.section) of newborn mouse epidermal whole mounts. **b**, **c**, **e**, **g** Representative image of *n* ≥ 3 biological replicates each. Nuclei were stained with DAPI (blue).

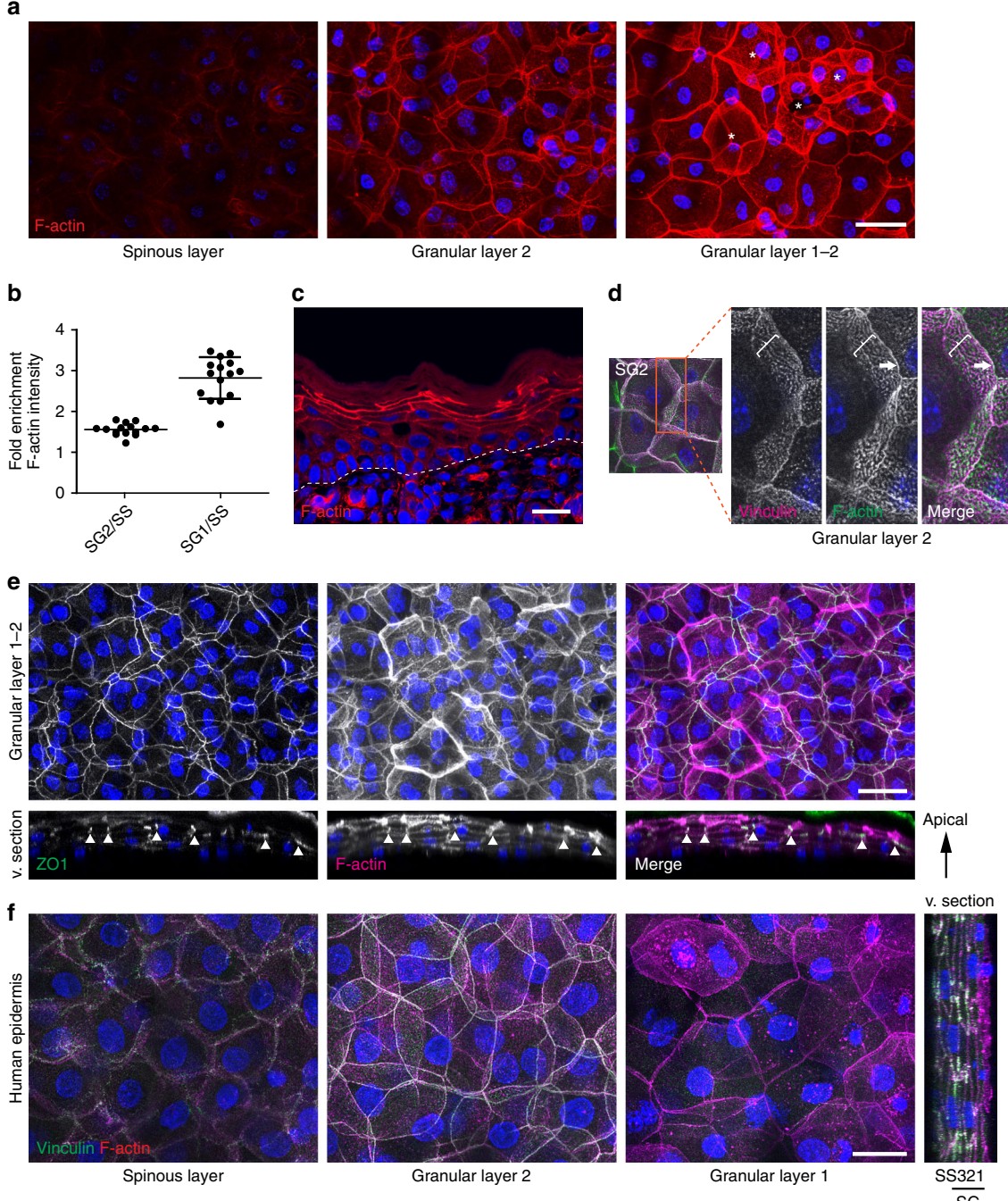

**Fig. 2** Highest F-actin organization in TJ containing granular layer. **a** Newborn epidermal whole-mount immunofluorescence analysis for Phalloidin revealing strong F-actin organization in granular layer 1 (SG1) and 2 (SG2). Asterisks mark F-actin[high] SG1 cells. Scale bar, 30 μm. **b** Quantification of relative F-actin distribution in spinous and granular layers of epidermal whole mounts confocal stacks of three biological replicates. Graph shows mean values ± SD as well as single measurements (dots, $n = 5$ per replicate). **c** Immunohistochemical analysis for F-actin (phalloidin) on newborn mouse epidermis sections. Scale bar, 20 μm. **d** F-actin organization in SG2 showing F-actin at the apical junction ring (arrows) and at the lateral, vinculin-positive AJ network (bracket). **e** Whole-mount analysis using Phalloidin and ZO-1 showing that F-actin[high] cells are in SG1 above the TJ barrier (ZO-1) (arrowheads). Scale bar, 30 μm. **f** Human adult epidermis whole-mount immunofluorescence analysis for F-actin (phalloidin) and vinculin showing tissue polarity of F-actin and vinculin and presence of F-actin[high] cells in SG1 layer. Scale bar, 20 μm. **a**, **d**–**f** Partial confocal stack projections and virtual sections (v. section) from newborn mouse epidermal whole mounts **a**, **d**, **e** and human adult epidermis (**f**). **a**, **c**–**e** Representative image of $N \geq 3$ biological replicates each. **f** Representative image of two individuals. Nuclei were stained with DAPI (blue)

even though many of the molecular players required for barrier formation are shared between simple and multi-layered epithelia[2], the mechanisms that drive this tissue polarity are unknown.

The self-renewing epidermis balances proliferation in the basal layer with a tightly controlled differentiation program where cells move upward while undergoing stepwise transcriptional and cell shape changes to form the distinct suprabasal layers (Fig. 1a): the stratum spinosum (SS), stratum granulosum (SG), and stratum corneum (SC). As barrier function is preserved despite dynamic turnover, the changes in cell shape and position require active

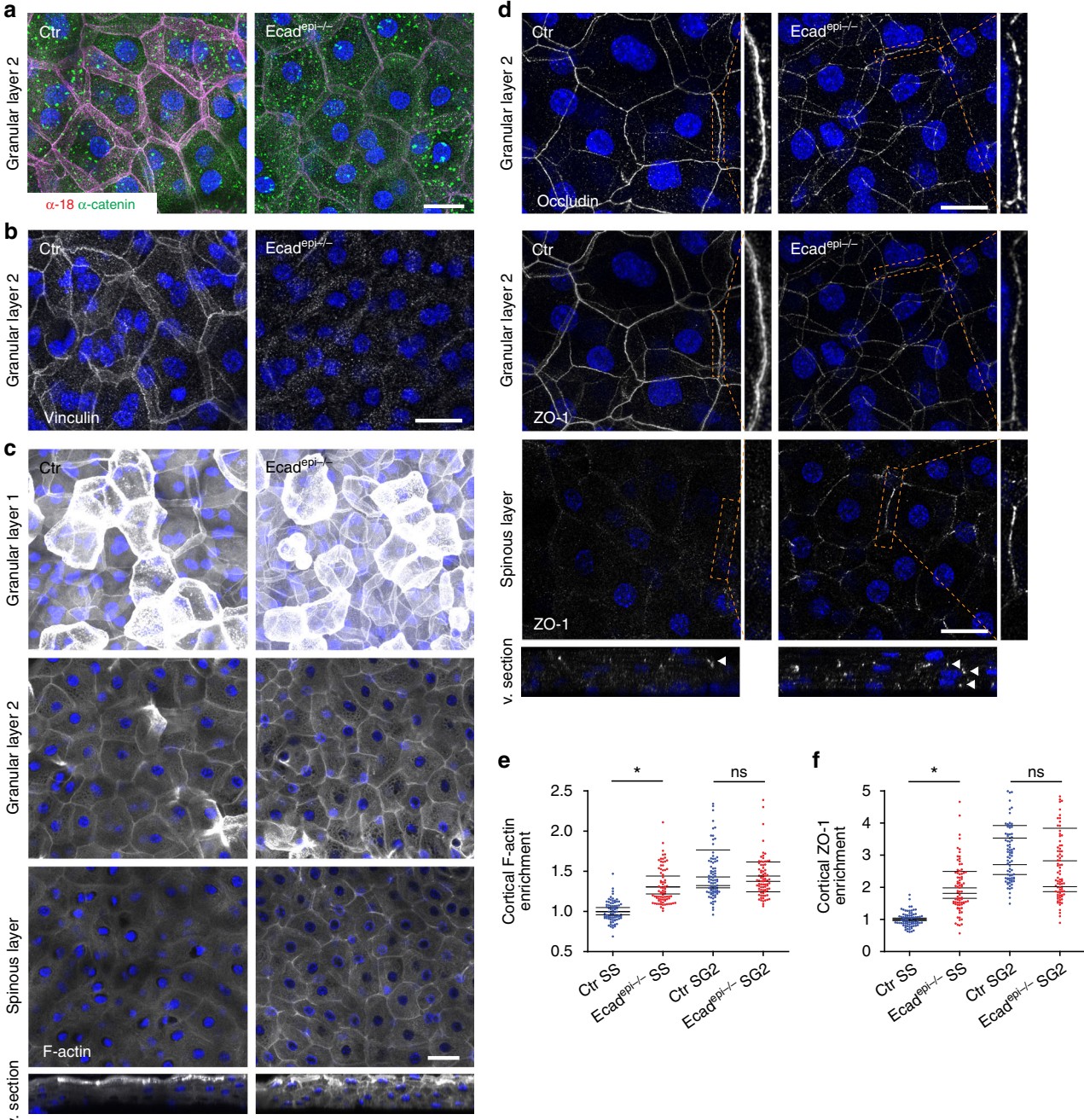

**Fig. 3** E-cadherin regulates tissue polarization of mechanosensitive junctions and cytoskeleton. **a** Newborn epidermal whole-mount immunofluorescence analysis for tension-sensitive epitope (α18) and total α-catenin and **b** Vinculin showing loss of α18 and vinculin and strong reduction in total α-catenin in granular layer 2 (SG2) of Ecad^epi−/−. Scale bar, 20 μm. **c** F-actin staining (Phalloidin) showing that loss of E-cadherin results in less F-actin polarization across layers. Scale bar, 30 μm. **d** ZO-1 and occludin staining showing interrupted TJs in SG2 and increased punctate ZO-1 in lower layers upon loss of E-cadherin. Scale bar, 20 μm. **a**, **b**–**d** Partial confocal stack projections and virtual sections (v. sections) from newborn mouse epidermal whole mounts. Stacks shown in **a** were deconvolved. **a**, **b**–**d** Representative images of $n \geq 3$ biological replicates each. Nuclei were stained with DAPI (blue). **e** Quantification of F-actin intensities from **c** normalized to Ctr spinous layer (SS). 20 measurements (dots) per biological replicate and the mean (lines) of each biological replicate are shown, *$P = 0.0308$; $n = 4$ biological replicates with Kruskal–Wallis, Dunn's post hoc test. **f** Quantification of layer specific cortical ZO-1 enrichment normalized to Ctr SS. 20 measurements (dots) and the respective means (lines) of four biological replicates are shown, ns = not significant, *$P = 0.0286$; $n = 4$ biological replicates with Mann–Whitney test

and tightly coordinated rearrangements of intercellular junctions. Improper development or maintenance of the epidermal barrier results in a range of skin diseases including very common inflammatory skin diseases and skin cancer[3, 4]. It is thus of fundamental importance to understand how the epidermal barrier, including the TJ barrier present in the SG, is formed and maintained.

Classical cadherins such as E-cadherin and P-cadherin are key adhesive components of AJs that couple intercellular adhesion to the cytoskeleton via β-catenin and α-catenin[5]. The latter can directly interact with actin but also binds a range of actin binding proteins. Recently, it has become clear that classical cadherins not only withstand tension but also sense mechanical forces and transduce this tension across cells in an actomyosin-dependent

manner to expand junctional size and complexity[6]. This mechanotransduction provides increased stiffness and mechanical stability to intercellular contacts, cells, and tissues. On a molecular level, increased tension across junctions results in the unfolding of α-catenin, exposing a binding site for vinculin, which then reinforces interactions with the actomyosin cytoskeleton[7, 8].

Studies in lower organisms and mammalian cell cultures have identified a central role for E-cadherin in coordinating the establishment of apico–basolateral polarity with the formation of AJs, desmosomes, and TJs[1, 9]. However, whether and how cadherins control tissue tension and polarized organization of junctions and the actin cytoskeleton in mammalian epithelial tissues in vivo is less clear. In the epidermis loss of E-cadherin expressed in all viable layers interferes with the TJ barrier, resulting in perinatal death[10]. This raises the intriguing question of how the ubiquitously localized E-cadherin controls formation of TJs only in the upper viable layer.

In the present study, we combine whole-mount imaging, epidermal-specific genetic ablations in vivo, and traction force microscopy (TFM) and atomic force microscopy (AFM) to address this question. We show that E-cadherin is a master regulator of junctional and cytoskeletal tissue polarity in stratifying epithelia, where it coordinates positioning of vinculin-positive tension-bearing AJs, barrier-forming TJs, and cortical F-actin to the uppermost viable layers of the epidermis. Mechanistically, we find that this coordination is driven by E-cadherin-dependent control of tension that spatiotemporally regulates the suprabasal localization and activation status of the epidermal growth factor receptor (EGFR) essential to facilitate the development of a functional epidermal barrier.

## Results

**Tissue polarization of junctions**. To examine the spatial relationship between AJs and TJs on the tissue level and within individual cell layers (Fig. 1a), we overcame the limited resolution of traditional skin tissue sagittal sections by adapting an epidermal whole-mount technique[11] to newborn mouse epidermis and performed high-resolution microscopy. As cadherin-mediated mechanotransduction through vinculin has been implicated in TJ barrier function of simple epithelia[12], we first asked how indicators of cadherin-mediated mechanotransduction are organized across epidermal layers. Staining for E-cadherin confirmed the presence of E-cadherin-containing AJs at sites of cell–cell contacts in all layers, with no apparent enrichment in the granular layer, where TJs are formed (Fig. 1b, e, f). Surprisingly, vinculin, which is recruited to AJs upon mechanosensitive unfolding of α-catenin[8, 13], exhibited a polarized tissue distribution and was highly enriched (4-fold) at intercellular contact sites in the second granular layer (SG2, Fig. 1a, b, f; Supplementary Fig. 1a). Here it co-localized with E-cadherin (Fig. 1b). Staining with α18-antibody (AB) that detects a mechanosensitive epitope in α-catenin[13], further indicated that these junctions were under high mechanical tension (Fig. 1c, d; Supplementary Fig. 1b).

We next examined the architecture of TJs that are exclusively formed in the SG2 layer[10, 14, 15]. The TJ marker ZO-1 is found at intercellular membranes in all layers, but, as previously reported[16, 17], ZO-1 showed an enrichment at intercellular contacts of the SG2 layer (Fig. 1e, f). Closer examination revealed that SG2 cells formed an E-cadherin/vinculin-rich basolateral network of AJs, which at its most apical border partially overlapped with the apically localized ZO-1-positive TJ network. Thus, despite their flattened shape, SG2 cells still exhibit junctional cell polarity (Fig. 1g; Supplementary Movie 1). Together the data show that, despite homogenous E-cadherin

distribution, AJs display differential molecular compositions along the basal-to-apical tissue axis. Within the SG2 vinculin-positive AJs form a basolateral network that support the apically localized TJs exclusively formed in this layer.

**Differential F-actin organization across layers**. As mechanical reinforcement of AJs and of functional TJs appears to be restricted to the SG2, we next addressed how these junctions relate to the epidermal organization of the F-actin cytoskeleton. Previous findings using K14-actin-GFP transgenic mice indicated a relatively evenly distributed cortical actin network[18]. However, actin-GFP marks both G-actin and F-actin and may also interfere with actin dynamics[19]. Phalloidin staining of whole mounts and tissue sections revealed a highly organized cortical F-actin network in the SG2 layer, whereas cortical staining was much less pronounced in the spinous layer (Fig. 2a–c). Within the SG2 layer, F-actin associated with both the apical positioned linear TJ junctions and the basolateral vinculin-positive AJ networks (Fig. 2d). Interestingly, the strongest F-actin staining was found in scattered cells in the SG1 granular layer above the TJs (Fig. 2a, asterisks, b, e; Supplementary Movie 2). Importantly, detection of F-actin using Lifeact-GFP transgenic mice revealed a similarly polarized distribution for F-actin including occurrence of scattered F-actin[high] cells in SG1 (Supplementary Fig. 2a, asterisks), excluding that reduced phalloidin staining in lower layers resulted from epitope masking or insufficient permeabilization. Staining for phosphorylated myosin (pMLC) showed strongest intensity also within the granular layer (Supplementary Fig. 2c), indicating increased actomyosin contractility in this layer. Polarized tissue distribution of vinculin-positive junctions and F-actin was also observed in human adult whole-mount epidermis (Fig. 2f; Supplementary Fig. 2b), demonstrating that tissue asymmetry for junctions and cytoskeleton is a general phenomenon of stratifying epithelia.

**E-cadherin controls junctional tissue polarization**. We next addressed how polarized organization of junctional tension and F-actin along the basal-to-apical tissue axis is achieved. As E-cadherin controls apico–basolateral polarity of simple epithelia[1] and controls TJ function in the epidermis[10], we hypothesized that E-cadherin could mediate tissue-level polarity. Whole-mount analysis revealed a strong reduction in both total α-catenin in suprabasal layers and the mechanosensitive α18 epitope in the SG2 layer in E-cadherin-deficient epidermis (from here on Ecad[epi−/−])(Fig. 3a). Furthermore, junctional vinculin was lost in the granular layer (Fig. 3b; Supplementary Fig. 3a), albeit with no apparent change in total protein level (Supplementary Fig. 3b). Thus, E-cadherin-dependent AJs are the predominant vinculin-positive, tension-bearing AJs in suprabasal epidermal layers.

We next asked whether E-cadherin loss would affect the asymmetric tissue distribution of F-actin across epithelial layers. Surprisingly, despite absence of AJs in the SG2 layer, phalloidin staining of whole mounts revealed no obvious changes in F-actin organization neither in the SG2 nor SG1 layer (Fig. 3c, e). In contrast, the spinous layers showed an increase in cortical F-actin organization (Fig. 3c, e; Supplementary Fig. 3c, d). Moreover, the TJ markers ZO-1 and occludin lost their continuous and linear apical distribution in Ecad[epi−/−] SG2. Instead, these proteins showed a more punctate pattern at cellular interfaces, revealing many breaks (Fig. 3d), thus explaining why in vivo TJ barrier function is disturbed upon loss of E-cadherin[10]. Interestingly, this punctate intercellular contact pattern of ZO-1 and occludin was also observed in Ecad[epi−/−] spinous layers, indicating premature formation of non-functional TJ-like structures (Fig. 3d, f; Supplementary Fig. 3e; ref. [10]). Thus, E-cadherin controls basal-

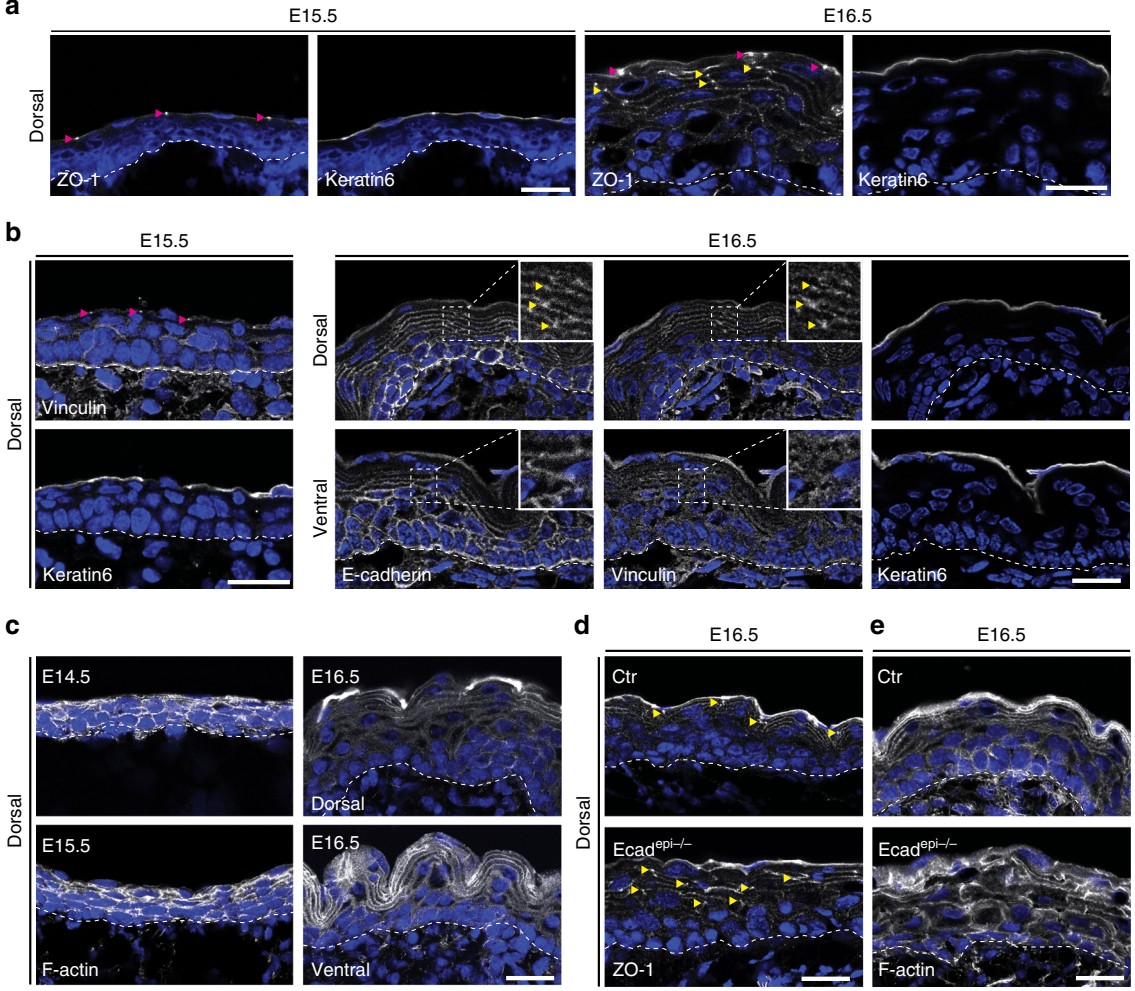

**Fig. 4** E-cadherin spatiotemporally coordinates the formation of tension-high adherens junctions, tight junctions, F-actin, and barrier function. **a–e** Immunohistochemistry analysis on mouse embryonic skin before (E15.5, E16.5 ventral) and after (E16.5, dorsal) epidermal barrier formation for **a** TJ marker ZO-1 showing that TJ formation in uppermost layer (yellow arrowheads) coincides with barrier formation. Peridermal TJs are marked by magenta arrowheads. **b** Staining for vinculin showing that suprabasal, junctional recruitment (yellow arrowheads) coincides with barrier formation. Keratin-6 was used to identify periderm (**a**, **b**). **c** Phalloidin staining showing F-actin organization and polarization. **d**, **e**. Staining for ZO-1 **d** and Phalloidin **e** showing increased depolarization of ZO-1 junctional recruitment and F-actin organization upon loss of E-cadherin during barrier formation. **a–c** Representative image of $n \geq 3$ biological replicates each. **d**, **e** Representative image of $n = 2$ biological replicates each. Nuclei were stained with DAPI (blue). Scale bars, 20 μm

to-apical tissue organization of vinculin-positive AJs and F-actin across epidermal layers and regulates the positioning of functional TJs to the SG2 layer.

**Spatiotemporal coordination of F-actin and barrier formation**. We next asked when TJs are formed during morphogenesis. A functional SC barrier is first formed in dorsal epidermis at embryonic day (E)16.5, spreading only to the ventral side one day later[10, 20]. Interestingly, in E14.5 and E15.5, ZO-1 was absent from intercellular epidermal contacts, and only stained TJs of the periderm, a simple provisional barrier-forming epithelium (Fig. 4a; Supplementary Fig. 4a). Similarly, little junctional vinculin was observed in E15.5 epidermis or in E16.5 ventral epidermis, prior to SC formation, despite uniform distribution of E-cadherin at all intercellular contacts in both E16.5 ventral and dorsal epidermis (Fig. 4b). At E16.5, the onset of a functional SC, ZO-1 became strongly enriched at junctions of the uppermost viable dorsal epidermal layer, together with vinculin (Fig. 4a, b), even though total vinculin remained similar (Supplementary Fig. 4b, c). Similarly, a basal-to-apical polarization of F-actin

became only obvious in E16.5 dorsal epidermis, with cortical F-actin now being enriched in the granular layer (Fig. 4c) in contrast to uniform F-actin in E14.5/15.5 embryos. E16.5 ventral regions showed only partially initiated suprabasal F-actin polarization, which was not yet confined to the granular layer (Fig. 4c).

Importantly, loss of E-cadherin interfered with the tissue polarization of ZO-1 and F-actin in dorsal E16.5 epidermis, resulting in a more uniform distribution across all layers (Fig. 4d, e). Thus, E-cadherin directs the polarized tissue distribution of junction tension, F-actin, and TJs at E16.5, which coincides with initiation of functional SC barrier activity.

**E-cadherin coordinates intercellular forces**. To assess whether E-cadherin directly controls junctional tension and actomyosin organization we isolated primary keratinocytes of control and Ecad$^{epi-/-}$ mice and switched these cells from low to high Ca$^{2+}$-concentration to initiate intercellular junction formation. Control cells formed regularly spaced cadherin-catenin-complex-positive early intercellular junctions, also known as zippers[21], which recruit both vinculin, F-actin and ZO-1 (Fig. 5a–c;

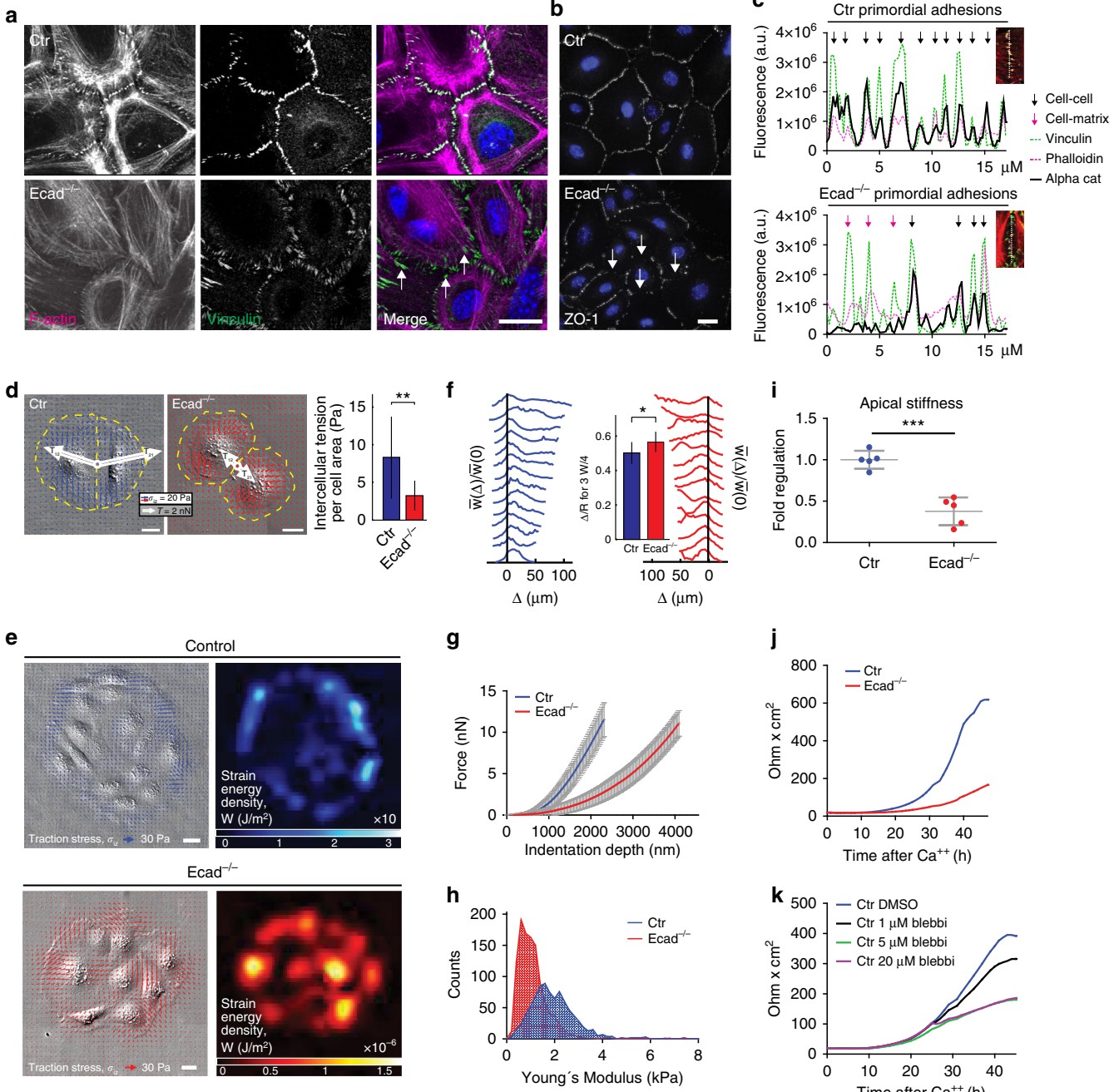

**Fig. 5** E-cadherin controls AJ zipper formation and intercellular tension. **a** Immunofluorescence analysis for vinculin and F-actin and **b** ZO-1 in keratinocytes 2 h in high $Ca^{2+}$. White arrows indicate cell–matrix contacts **a** and irregular zipper formation **b**. **c** Line plot profile of cell contacts, pink arrows: cell–matrix contacts, black arrows: cell–cell contacts. **a–c** Representative images/profiles of $n \geq 3$ biological replicates. **d** Traction force microscopy. (Left) Distributions of in-plane traction stresses (blue and red arrows), $\sigma_{iz}$, for pairs of control (Ctr) or E-cadherin (Ecad)$^{-/-}$ keratinocytes after 24 h in $Ca^{2+}$, overlaid on DIC images. White arrows: resultant intercellular tension, $T_{12}$, of cell 2 on cell 1, and vice versa, given by vector sum of in-plane tractions within dashed yellow boundary of opposite cell of the pair. (Right) Quantification of intercellular tensions per cell area for Ctr ($n = 14$) or Ecad$^{-/-}$ ($n = 15$) keratinocyte pairs. Error bars: standard deviations. **$P = 0.002$, Student's $t$-test. **e** In-plane traction stresses (blue and red arrows) overlaid on DIC images and strain energy densities (blue and red heat maps) of Ctr and Ecad$^{-/-}$ keratinocyte colonies 24 h in high $Ca^{2+}$. **f** Line plots of normalized strain energy distributions from edge ($\Delta = 0$) to colony interior for Ctr (blue, $n = 14$) and Ecad$^{-/-}$ (red, $n = 14$) colonies. **f** Graph represents fraction of inward displacement from colony edge needed to capture 3/4 of total strain energy. Error bars: standard deviations. *$P = 0.0102$, Student's $t$-test. Scale bars, 20 μm. **g** AFM force indentation plots, mean plots of $n = 8$ representative measurements including 2 primary Ctr and E-cad$^{-/-}$ cell lines after 48 h in high $Ca^{2+}$ are shown. **h** Distribution of Young's moduli from a representative indentation experiment on 2 primary Ctr and E-cad$^{-/-}$ cell lines 48 h in high $Ca^{2+}$ (Ctr: $n = 853$, E-cad$^{-/-}$: $n = 1099$). **i** Relative Young's moduli in Ctr and E-cad$^{-/-}$ cells. Median values of biological replicates were used for statistical analysis. ***$P = 0.0001$; Ctr/Ecad$^{-/-}$ $n = 5$ biological replicates with Student's $t$-test (>300 measurements/replicate). **j** Transepithelial resistance (TER) measurements in Ctr and Ecad$^{-/-}$ keratinocytes. Representative example of $n > 10$ independent isolates. **k** TER measurements in primary keratinocytes upon Blebbistatin treatment 24 h after switching to high $Ca^{2+}$. Scale bars, 20 μm

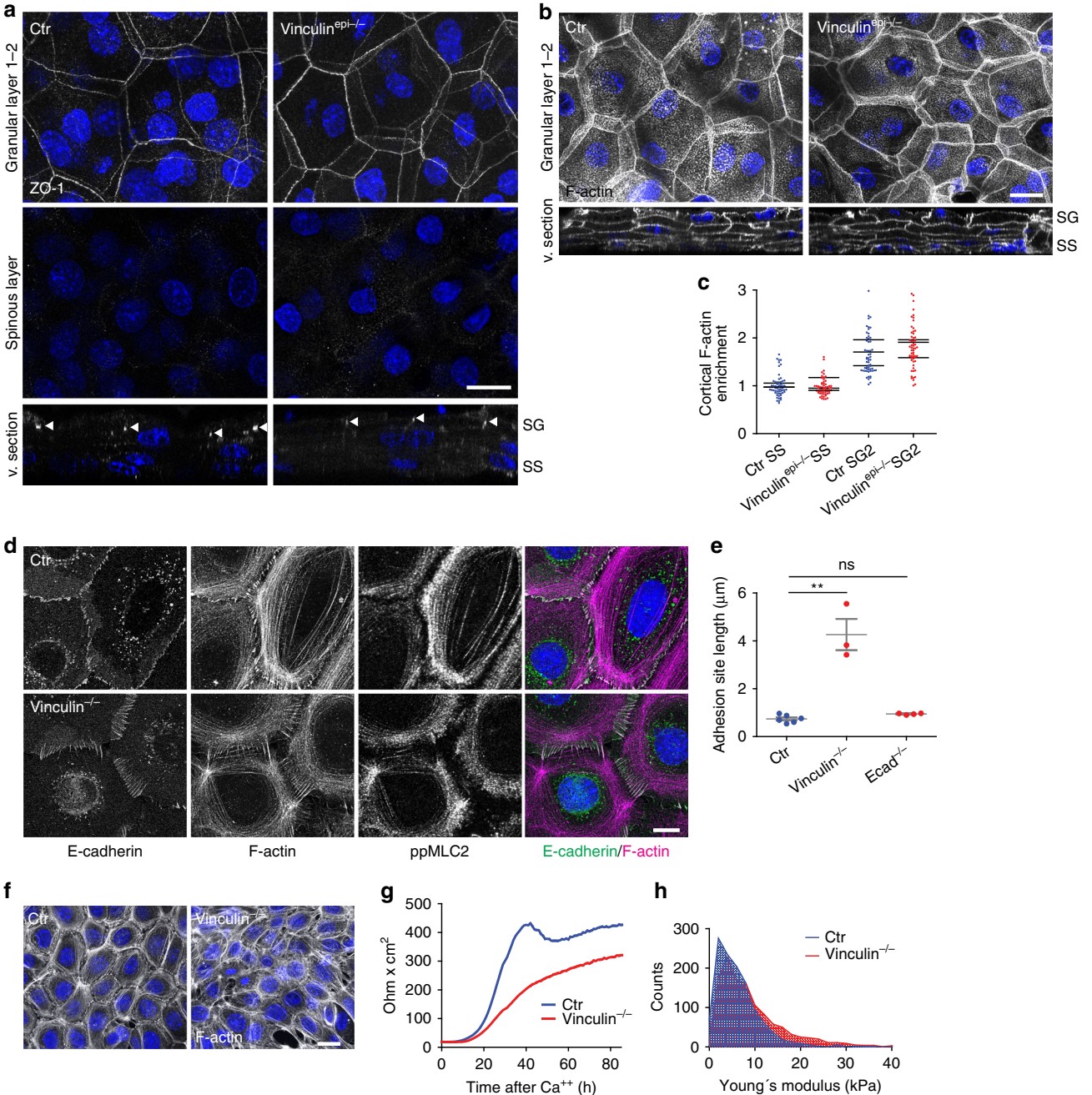

**Fig. 6** Vinculin is not essential for junctional tissue polarization. **a** Newborn epidermal whole-mount fluorescence analysis for ZO-1 showing its distribution in granular layer 2 (SG2), spinous layer (SS), and cross section (v. section). **b** Newborn epidermal whole-mount fluorescence analysis for phalloidin showing F-actin organization in SG2 and in all epidermal layers (v. section). **a**, **b** Partial confocal stack (deconvolved) projections and virtual section (v. section). Nuclei stained with DAPI (blue). Scale bars, 20 μm. **c** Quantification of layer specific cortical actin enrichment normalized to Ctr SS, showing no change upon loss of vinculin. **d** Immunofluorescence analysis for E-cadherin, F-actin (phalloidin) and active myosin (ppMLC2) in primary control and vinculin$^{-/-}$ keratinocytes after 2 h in high Ca$^{2+}$. Scale bar, 10 μm. **e** Quantification of AJ length. **$P < 0.05$; $n = 6$ Ctr, $n = 3$ vinculin$^{-/-}$, $n = 4$ E-cadherin$^{-/-}$ biological replicates ($n = 100$ AJs per replicate) with Kruskal–Wallis, Dunn's post hoc test. **f** Low magnification images of F-actin organization corresponding to **d**. Scale bar, 20 μm. **d**, **f** Representative image of $n \geq 3$ independent keratinocyte isolations. **g** Transepithelial resistance measurements in primary Ctr and vinculin$^{-/-}$ keratinocytes. Representative example of $n = 4$ biological replicates. **h** Histogram showing the distribution of Young's moduli obtained from a representative indentation experiment on 3 primary Ctr and 3 vinculin$^{-/-}$ cell lines after 48 h in high Ca$^{2+}$ (Ctr: $n = 1194$, vinculin$^{-/-}$: $n = 1207$ measurements)

Supplementary Fig. 5a). E-cadherin$^{-/-}$ keratinocytes showed strong impairment in early zipper formation, with most of the vinculin localized to focal contacts at the cell–matrix interphase (Fig. 5a, c; Supplementary Fig. 5a, white arrows). Importantly, TFM on doublets of cells[22] revealed a 3-fold reduction in tension across intercellular contacts in E-cadherin$^{-/-}$ keratinocytes compared to control after 24 h in high Ca$^{2+}$ (Fig. 5d). This was surprising as desmosomes, considered an important mechanical unit, are formed in these cells[10, 23]. Moreover, loss of E-cadherin interfered with coordinated transfer of force across multiple cells to cell–matrix adhesion sites at edges of multicellular colonies (Fig. 5e, f)[24]. These alterations in mechanical behavior were not

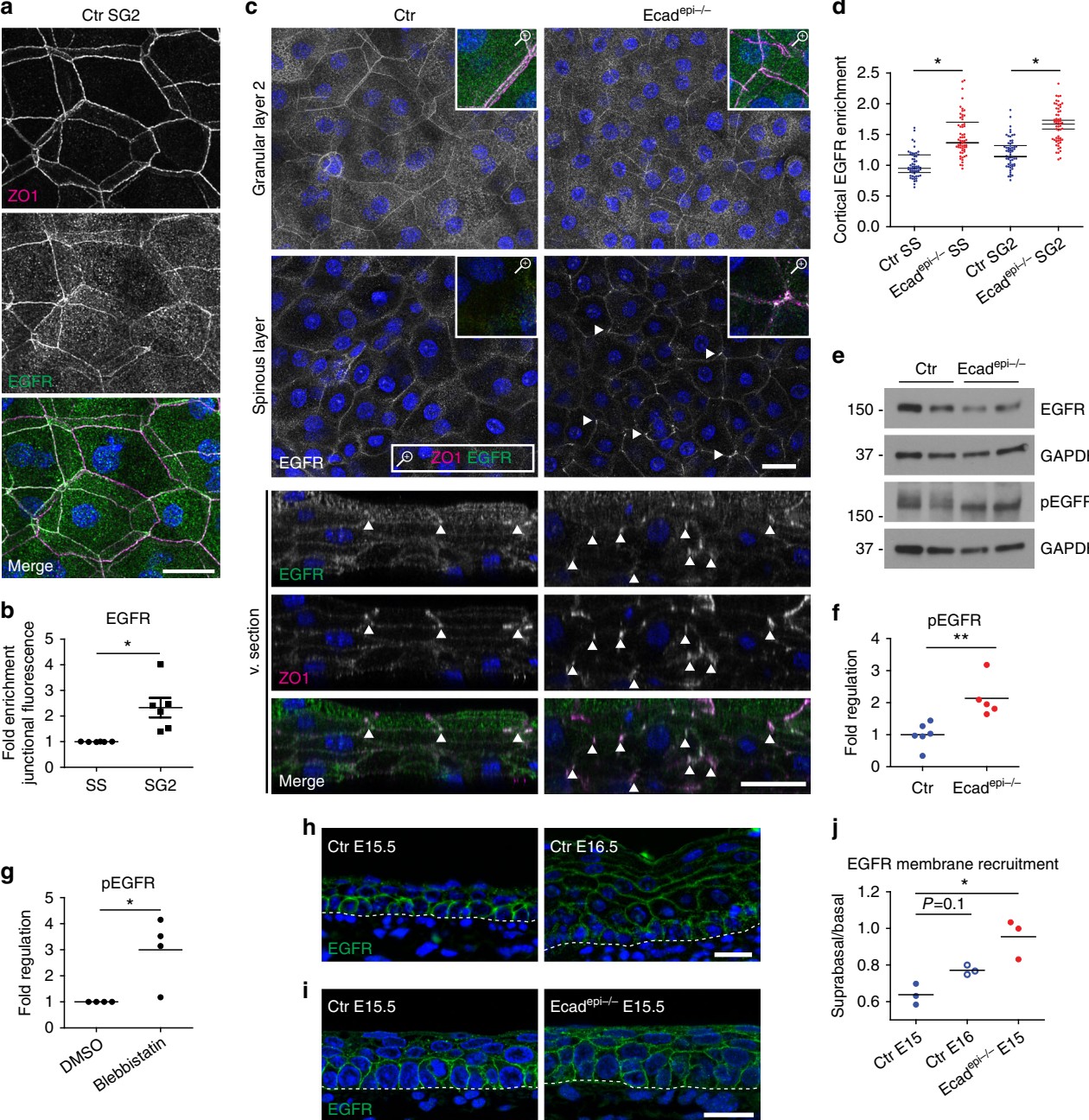

**Fig. 7** E-cadherin regulates EGFR tissue polarization and activity. **a** Newborn epidermal whole-mount fluorescence analysis for EGFR and ZO-1 showing enrichment of EGFR at TJs in control SG2 layer. **b** Quantification of junctional EGFR fluorescence intensity. $n = 6$ biological replicates with 10 junctions each. Graph shows mean values ± SEM and single means of biological replicates (dots/squares). Granular layer (SG) intensity was normalized to spinous layer (SS) intensity. *$P = 0.0214$ with D'Agostino & Pearson omnibus normality test followed by one sample $t$-test. **c** Newborn epidermal whole-mount fluorescence analysis for EGFR and ZO-1 showing SG2, SS, and cross section (v. section), showing increased junctional ZO-1 and EGFR (arrowheads) in the SS of E-cadherin (Ecad)$^{-/-}$ epidermis compared to control (Ctr). **a**, **c** Partial projections and v. sections of confocal stacks from newborn epidermal whole mounts. For better visualization fluorescence intensities of SS projections have been increased relative to SG. Nuclei were stained with DAPI (blue). **d** Quantification of cortical EGFR enrichment in SG normalized to Ctr SS. Shown are respective means (lines) of three biological replicates with 20 measurements (dots) per replicate, *$P < 0.05$; $n = 3$ biological replicates with Kruskal–Wallis, Dunn's post hoc test. **e** Western blot for total and phosphorylated EGFR (pEGFR) from Ctr and Ecad$^{epi-/-}$ newborn epidermis. **f** Western blot quantification of pEGFR levels in newborn Ctr and Ecad$^{epi-/-}$ epidermis. **$P = 0.0042$; Ctr $n = 6$, Ecad$^{epi-/-}$ $n = 5$ biological replicates with Student's $t$-test. **g** Quantification of western blot for phospho-EGFR (72 h in high Ca$^{2+}$, blebbistatin added after 24 h). *$P < 0.05$; $n = 4$ biological replicates with one sample $t$-test, hypothetical value = 1 (normalized to control treated DMSO). **h** Staining of embryonic skin for EGFR before (E15.5) and after epidermal barrier formation (E16.5, dorsal skin). **i** Immunofluoresence analysis of E15.5 control and Ecad$^{epi-/-}$ skin for EGFR shows premature suprabasal localization of EGFR. Nuclei were stained with DAPI (blue). **j** Quantification of cell membrane EGFR intensities (mean gray value) in suprabasal vs. basal layers. The ratio of suprabasal vs. basal layer membrane intensity is plotted. Dots represent means of biological replicates. *$P < 0.05$; $n = 3$ biological replicates each group with Kruskal–Wallis, Dunn's post hoc test. Scale bars, 20 μm

accompanied by changes in ppMLC2 levels or localization (Supplementary Fig. 5b–d). This inability to mechanically couple cells through junctional zippers was neither due to reduced vinculin expression (Supplementary Fig. 5e, f) nor to a failure of P-cadherin, the other epidermal classical cadherin, to sense and respond to mechanical signals, as vinculin and F-actin were still recruited to individual AJs in E-cadherin$^{-/-}$ keratinocytes (Fig. 5c; Supplementary Fig. 5a). Overexpression of either P-cadherin or E-cadherin also rescued the formation of regularly spaced vinculin-positive AJ zippers (Supplementary Fig. 5g, h). Together, these results indicate that a threshold expression level of classical cadherins is necessary to coordinate the organization of early tension-bearing junctions into a regularly spaced zipper to coordinate force transduction across cells.

**E-cadherin coordinates TJ barrier through actomyosin.** As both tension-bearing AJs and TJ formation were restricted to the suprabasal SG2 layer in vivo (Fig. 1), which does not contain tension-bearing cell–matrix contacts, we next asked whether E-cadherin would specifically regulate mechanical properties of suprabasal cells. When primary keratinocytes are exposed to high Ca$^{2+}$ for 24–72 h, cells stratify and TJs become localized to the suprabasal layer of a multi-layered epithelial sheet[18]. As the three-dimensional nature of this multi-layered structure where forces are dissipated in multiple directions precludes laser ablation or TFM, we quantified cell mechanics in this layer using force indentation spectroscopy with an AFM. Using blebbistatin, we first confirmed that the cortical actomyosin network controls the measured elastic properties of these cells (Supplementary Fig. 5i). Interestingly, AFM measurements showed a reduction in cortical stiffness upon loss of E-cadherin (Fig. 5g–i) but not upon knockdown of the TJ protein ZO-1 (Supplementary Fig. 5j). These alterations in cell mechanics were accompanied by the inability of E-cadherin$^{-/-}$ keratinocytes to establish a functional barrier, as measured by transepithelial resistance (TER) measurements (Fig. 5j). In contrast, control keratinocytes efficiently developed a functional barrier over time, showing that E-cadherin-mediated barrier formation is intrinsic and does not require a SC. Importantly, inhibiting myosin activity interfered with barrier formation in control keratinocytes (Fig. 5k), showing that actomyosin tension directly controls the TJ barrier. Together, these results indicate that proper coupling of E-cadherin-containing AJs to the actomyosin cytoskeleton is crucial to couple and coordinate tension across a stratifying epithelial sheet essential to establish and reinforce the barrier.

**Vinculin not required for junctional tissue polarization.** We next addressed the role of vinculin recruitment to AJs in early AJ zipper formation and in vivo epidermal polarization of junctions and the cytoskeleton. To this end, we generated vinculin-floxed mice (Supplementary Fig. 6a, b) and crossed them with K14-Cre mice[25] to delete vinculin in the epidermis (Supplementary Fig. 6c). In vivo, loss of epidermal vinculin did not result in perinatal lethality unlike loss of E-cadherin[10]. Epidermal whole mounts from vinculin$^{epi-/-}$ mice revealed no obvious defects in polarized ZO-1 tissue distribution or in its linear apical TJ localization within SG2 (Fig. 6a). More surprisingly, junctional actin in the granular layer or actin polarization along the basal-to-apical tissue axis were both not obviously affected by the loss of vinculin (Fig. 6b, c).

We then assessed whether vinculin was essential for initial adhesion zipper formation. Primary vinculin$^{-/-}$ keratinocytes were able to organize E-cadherin-ZO-1 positive junctions into regular zipper-like structures that recruited F-actin but not paxillin (Fig. 6d, f; Supplementary Fig. 6d–f), further confirming

that these are cell–cell and not cell–matrix junctions. However, these junctions were strongly elongated compared to those in control or E-cadherin$^{-/-}$ cells (Fig. 6d, e). Moreover, unlike in controls, F-actin was still predominantly organized in stress fibers, indicating that vinculin is necessary for early transition of F-actin from stress fibers into a cortical organization (Fig. 6f). In agreement, staining for ppMLC2 showed a more diffuse pattern compared to control (Fig. 6d). Thus, vinculin controls the length of the individual primordial AJs as well as overall cortical organization of the F-actin cytoskeleton. However, unlike E-cadherin deficiency, vinculin loss does not obviously impair the regularly spaced organization of early AJs into zippers (Fig. 5b; Supplementary Fig. 6e). Although TER measurements showed a reduced TJ barrier function upon loss of vinculin (Fig. 6g), this reduction was much less severe than that induced by loss of E-cadherin (Fig. 5j). In agreement, loss of vinculin did not obviously affect cortical stiffness of the suprabasal keratinocyte sheet as measured by AFM (Fig. 6h; Supplementary Fig. 6g, h). Thus, vinculin is important for initial junctional organization, suggesting it promotes mechanical reinforcement of newly formed junctions. It is, however, dispensable for polarized tissue organization of junctions, the cytoskeleton, and tension across epithelial layers and contributes only to a small extent to proper epidermal barrier formation. Our results suggest that other mechanically relevant AJ-associated proteins either control barrier formation or compensate for the in vivo loss of vinculin.

**E-cadherin regulates EGFR localization and activity.** We proceeded to identify the molecular mechanism by which E-cadherin controls tissue-level organization of cell mechanics and TJ positioning. As EGFR activity status has been linked to intercellular junctional and actomyosin organization and function[26–29], we examined this receptor in more detail. Staining of control tissue sections showed localization of EGFR not only in the basal layer, as expected[30], but also in suprabasal layers (Supplementary Fig. 7a, b). Whole-mount analysis revealed that EGFR was enriched at or near the apical ZO-1-containing TJs in SG2 (Fig. 7a, b). EGFR was also cortically enriched in the SG2 of human epidermis (Supplementary Fig. 7c). Upon loss of E-cadherin, EGFR staining in the SG2 layer became more punctate in addition to a more prominent staining in the spinous layers, where it partially co-localized with ZO-1 (Fig. 7c, d). In vitro, EGFR localized to early AJs showing a regular zipper-like distribution, which was strongly reduced in E-cadherin$^{-/-}$ keratinocytes (Supplementary Fig. 7d). Loss of E-cadherin also resulted in increased EGFR activity both in vivo (Fig. 7e, f; Supplementary Fig. 9) and in vitro (Supplementary Fig. 7e, f). Interestingly, this increased activity was not associated with enhanced proliferation in vitro or in vivo (ref. [10]; Supplementary Fig. 7g). Importantly, reintroduction of full length E-cadherin reversed this increased EGFR activity in E-cadherin$^{-/-}$ keratinocytes to control levels (Supplementary Fig. 7h, i), showing that E-cadherin directly regulates EGFR. To address whether E-cadherin inhibits EGFR by promoting tension we treated cells with blebbistatin, resulting in increased EGFR activity, indicating that tension is important (Fig. 7g; Supplementary Fig. 7j). Interestingly, however, an E-cadherin mutant that cannot interact with β-catenin and thus α-catenin also rescued increased EGFR activity in E-cadherin$^{-/-}$ keratinocytes, suggesting that a direct connection of junctions to actin is not required for E-cadherin-dependent regulation of EGFR (Supplementary Fig. 7h, i).

We then examined the in vivo spatiotemporal relationship of EGFR localization, junctional tension asymmetry, and acquisition of barrier properties. Interestingly, during early stages of epidermal morphogenesis, EGFR was restricted to basal

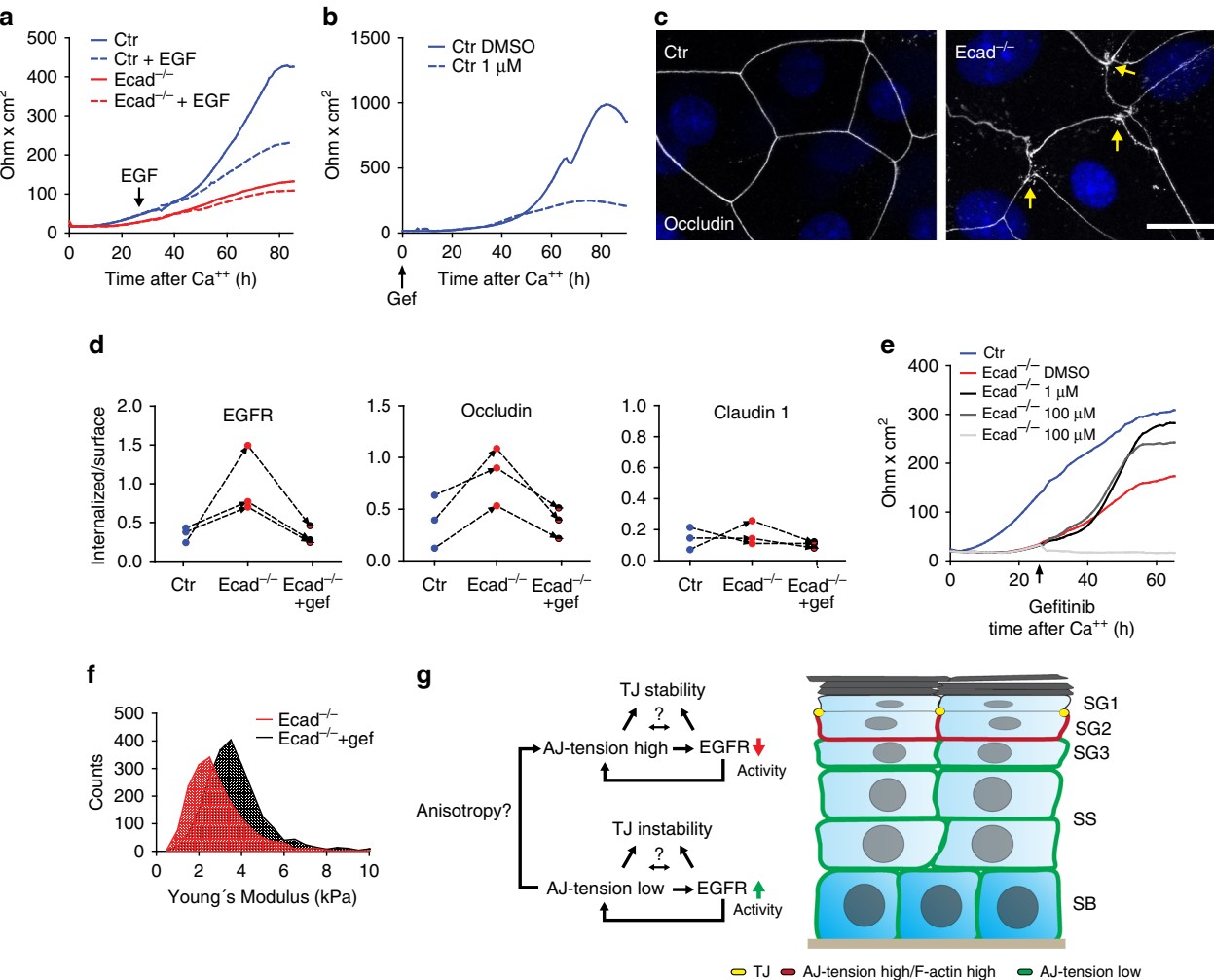

**Fig. 8** Spatiotemporal control of EGFR localization and activity controls TJ barrier function. **a**, **b** Transepithelial resistance (TER) measurements in primary keratinocytes. **a** Control (Ctr) keratinocytes were stimulated with EGF (100 ng ml$^{-1}$ final concentration) 24 h after switching to high Ca$^{2+}$ (dashed line). **b** Ctr keratinocytes were treated with EGFR inhibitor gefitinib starting with switching to high Ca$^{2+}$ (0 h). **c** Immunofluorescence analysis of occludin organization in Ctr and E-cadherin (Ecad)$^{-/-}$ primary keratinocytes after 48 h in high Ca$^{2+}$. Scale bar, 20 μm. Nuclei were stained with DAPI (blue). **d** Cell surface biotinylation/internalization assay. Graphs represent the relative amount of internalized protein per surface labeled protein 30 min after internalization. Indiviudal experiments are connected by dashed lines. **e** TER measurement, E-cadherin$^{-/-}$ keratinocytes were treated with different doses of EGFR inhibitor gefitinib and EGFR 24 h after switching to high Ca$^{2+}$. Note dose dependent rescue of TER in E-cadherin$^{-/-}$ keratinocytes. **a**, **b**, **f** Representative example of $n = 3$ biological replicates. **f** Histogram showing the distribution of Young's moduli obtained from a representative indentation experiment from primary either DMSO or gefitinib treated E-cadherin$^{-/-}$ keratinocytes after 48 h in high Ca$^{2+}$ (treatment after 24 h) ($n > 500$ measurements each treatment). **g** Model showing how E-cadherin integrates mechanical and chemical signals to restrict the formation of tight junctions to the stratum granulosum 2

intercellular contacts and was recruited to suprabasal junctions of dorsal epidermis only at E16.5 (Fig. 7h, j). This coincided with ZO-1 and vinculin recruitment to the most viable suprabasal layer (Fig. 4a, b) and with initiation of barrier function[20]. Most importantly, loss of E-cadherin resulted in premature recruitment of EGFR to suprabasal intercellular contacts at E15.5 (Fig. 7i, j). Thus, mislocalization of EGFR is the earliest change induced by loss of E-cadherin and precedes misguided localization of ZO-1 and actin in lower layers at E16.5 (Fig. 4d, e). In conclusion, E-cadherin controls the polarized localization and activity status of suprabasal EGFR, which is uncoupled from growth[10].

**EGFR activity controls apical stiffness and tight junctions.** As loss of epidermal EGFR is associated with skin barrier dysfunction[30], we next asked whether spatiotemporal regulation of EGFR activity would affect formation of TJ barrier function. EGF stimulation substantially lowered TER in control keratinocytes (Fig. 8a), but did not further reduce TER in E-cadherin$^{-/-}$ keratinocytes (Fig. 8a), consistent with the observation that EGFR activity is already increased in these cells. Vice versa, sustained inhibition of EGFR was sufficient to abrogate TJ barrier function (Fig. 8b). These findings show that either too little or too much EGFR activity interferes with formation of a functional TJ barrier.

We then asked whether enhanced EGFR activity induced by loss of E-cadherin directly controlled TJs. Immunofluoresence analysis revealed occludin-positive vesicle-like punctae in E-cadherin$^{-/-}$ keratinocytes but not in controls when cultured in conditions that allow for functional TJs (Fig. 8c). Quantitative cell surface biotinylation analysis revealed increased internalization of occludin and EGFR, but not of claudin-1, which was reversed upon inhibiting EGFR activity (Fig. 8d). Furthermore, inhibition of EGFR or its downstream mediator PKCα rescued TJ

barrier function (Fig. 8e; Supplementary Fig. 8a), but, interestingly, only when inhibitors were applied after junctions were allowed to form for 24 h in high calcium, a time point that coincides with stratification and initial barrier formation[31]. Importantly, EGFR inhibition also reversed the reduced cortical stiffness of E-cadherin$^{-/-}$ keratinocytes (Fig. 8f; Supplementary Fig. 8b), thus providing a direct link between chemical and mechanical signaling in the regulation of the barrier. Collectively, these data suggest that junctional tension generated by E-cadherin is required to localize and constrain EGFR activity, which in turn controls cortical stiffness and subsequent formation of TJs. Thus, E-cadherin-mediated spatiotemporal control of EGFR localization and activity is crucial for proper force transduction to correctly position the formation of a stable TJ barrier in multi-layered epithelia such as the epidermis (Fig. 8g).

## Discussion
The formation of epithelial tissue barriers is crucial for life and requires the coordinated positioning of junctions and cytoskeletal organization either along the apico–basolateral axis in simple epithelia or along the basal-to-apical tissue axis in multi-layered epithelia. The mechanisms underlying junctional positioning and coupling to the contractile actomyosin network to provide tensile strength to the barrier have been mostly explored in simple epithelia in lower organisms or in cell culture[32, 33]. Strikingly, virtually nothing is known how multi-layered epithelia such as skin epidermis establish and restrict junctional barrier function to the appropriate tissue layer. Here we unraveled an E-cadherin–EGFR-driven mechanical signaling network that spatiotemporally restricts intercellular tension and junction formation to allow robust assembly of the barrier only where it is spatially required, in the SG2 layer of the epidermis. The data also demonstrate a novel, and proliferation-independent function for suprabasal EGFR in the regulation of epidermal barrier formation.

Our results reveal striking similarities between junctional apico–basolateral polarity in simple epithelia and the polarized tissue organization of junctions in the epidermis. For example, E-cadherin is a key determinant of junctional polarity in both simple and stratified epithelia[9, 34], and apical positioning of tension-high, F-actin-rich junctions is seen in both types of epithelia; along the basal–apical axis of simple epithelial cells[35], or along the tissue axis in epidermis. Furthermore, cell-level apical–basolateral polarity is also observed within the epidermal SG2 layer, with the TJs being apically positioned from the AJs, thus resembling the apical positioning of simple epithelial TJs above the zonula adherens (ZA), the specialized tension-high AJs that are part of the apical junctional complex in simple epithelia. Nevertheless, crucial differences exist between simple and stratified epithelia that are likely to be directly coupled to fundamental differences in tissue function. Although SG2 junctions show structural resemblance to the tensile lateral networks recently described in cultured simple epithelia[36], they function more like the ZA in providing a continuous tensile network that mechanically couples contractile forces across epidermal sheets to promote barrier-forming TJs. Interestingly, in simple epithelia the ZA is the more contractile and stable junctional unit that apically borders the lateral AJ network and is necessary to prevent extrusion of cells from the monolayer[36]. In contrast, epidermal keratinocytes require continuous upward extrusion in order to turn over and renew the tissue[37]. The SG2 AJ lateral network in the epidermis may thus integrate these two functions by generating tension across cells while simultaneously allowing controlled extrusion towards the SG1.

Earlier in vitro work indicated vinculin as a crucial effector of cadherin-mediated mechanotransduction[38–42]. Our results identify fundamentally different junctional and mechanical roles for vinculin vs. E-cadherin, both in vitro and in vivo. Despite strong phenotypes early in intercellular junction assembly, vinculin, unlike E-cadherin, is dispensable for junctional, mechanical and cytoskeletal tissue organization that allow proper positioning of TJ in vivo. In addition, whereas E-cadherin controls the number and regular spacing of initial junctions assembling into zipper-like structures, vinculin is only required for local F-actin reorganization at these early junctions but does not control zipper formation itself. In agreement, loss of vinculin results in a much less pronounced disturbance of TJ barrier formation than loss of E-cadherin. In simple epithelia, vinculin was shown to control apical positioning of the ZA[43]. In contrast, loss of vinculin in the epidermis did not affect either apical ZO-1 positioning or F-actin distribution in SG2 and did not result in major barrier defects in vivo, as vinculin$^{epi-/-}$ mice are viable, unlike Ecad$^{epi-/-}$ mice generated using the same K14-Cre line[10]. The observation that myosin-II-mediated junctional tension in the epidermis was shown to be required for TJ barrier function[44], suggests that other proteins downstream of the cadherin complex may compensate and/or contribute to control actomyosin and junctional tissue organization and thereby barrier formation.

Our results identify a key role for E-cadherin in: (1) regulation of actomyosin-dependent tension, (2) junctional and cytoskeletal tissue polarization (3) EGFR localization and activation, and (4) TJ positioning and barrier formation. Our data furthermore show that actomyosin cytoskeleton controls cortical stiffness, EGFR activation and TJ formation, whereas tight regulation of EGFR itself determines cortical stiffness as well as TJ stability and function. Collectively, these data suggest that junctional tension generated by E-cadherin is required to localize and constrain EGFR activity, which in turn is required for cortical reinforcement and subsequent formation of TJs. In line with our data, actomyosin-dependent forces also constrain EGFR activity in simple epithelia[29]. In apparent contrast, it was recently found that E-cadherin-dependent force transduction activates EGFR, resulting in integrin and cell–matrix interaction-dependent stiffening of cells[28]. However, this study used single cells in combination with coated beads whereas cells within the SG2 layer or the tightly interconnected apical layer of keratinocyte multi-layered sheets do not adhere through integrin based cell–matrix contacts. Cadherin engagement, force transduction, and the observed tight, spatiotemporal regulation of EGFR activity may thus be fundamentally different in the two systems.

How E-cadherin controls suprabasal EGFR localization and activation status is not clear. Our in vitro data provide indirect evidence that E-cadherin regulates EGFR in a tension-dependent manner that, however, appears independent of stable association with the actin cytoskeleton (Fig. 7g and Supplementary Fig. 7h, i). VE-cadherin and VEGFR2/3 can form a mechanosensitive complex in endothelial cells that controls VEGFR signaling in a shear-force-dependent manner[45]. Along the same lines, E-cadherin itself was shown to interact with and regulate EGFR in vitro[46–49]. One possible model may thus be that actomyosin-dependent strengthening of junctions promotes a more stable interaction of E-cadherin with EGFR, thus resulting in effective inhibition of EGFR only in the SG2 layer. Unfortunately, despite extensive attempts including mass spectrometry analysis failed thus far to reveal an interaction in keratinocytes. Moreover, EGFR concentrates at or near ZO-1-positive TJs in newborn epidermis, suggesting that EGFR might interact with TJ proteins at least after the barrier has formed, providing an alternative mechanism for recruitment. Understanding how EGFR functions as a mechanosensor to control the polarized organization of junctional tension and barrier formation and function will be an important avenue for future research.

The localization of EGFR at the TJ barrier, where its activity needs to be tightly controlled to drive and, likely, maintain barrier function, provides a novel perspective on why cancer patients treated with EGFR inhibitors develop skin rashes of unknown origin[50]. Epidermal inactivation of EGFR in mice showed that these rashes were a direct consequence of epidermal EGFR, which was attributed to chemokine-driven inflammation[51–53]. Our results now offer the exciting possibility that EGFR may serve a dual function at epidermal TJs, which integrates the maintenance and restoration of TJ barrier function with tight control of epidermal cytokine expression essential to restore tissue homeostasis upon insults to the barrier. Either too much or too little EGFR activity at TJs will impair barrier function and alter cytokine expression, both of which contribute to skin inflammatory skin diseases[3] as well as skin cancer[54]. We thus propose that cadherin-dependent regulation of suprabasal junctional EGFR is not only important to establish barrier function but will also be essential to regenerate the barrier and restore homeostasis upon barrier disruption.

How the ubiquitously expressed E-cadherin controls basal-to-apical tissue polarization of EGFR, tension-bearing AJs, TJs, and F-actin is still an open question. As initiation of differential EGFR, junctions and F-actin tissue organization coincides with the formation of a protein and lipid cross-linked functional SC barrier (Fig. 4), this might suggest that the stiffness of the SC provides a mechanical cue to drive formation of tension-high AJs and TJs in SG2. Alternatively, the differential distribution of AJs in SG2 cells vs. lower cells may be a key determinant: in contrast to cells from lower layers that connect through AJs at all intercellular interfaces, in SG2 cells AJs are only found at basal and lateral but not apical SG2–SG1 intercellular contacts, thus resulting in an anisotropy of tension distribution that allows formation of tension-high AJs and TJs. As the in vitro multi-layered epithelial sheets do not form an SC but still restrict TJs to the most apical layer (Fig. 5j), this would favor the latter hypothesis. As AJs crosstalk with desmosomes that are coupled to keratins[23], and desmosome composition may also be different between granular and lower layers, it will be very interesting to address how desmosomes and keratins contribute to tissue polarization of junctions, cytoskeleton, and tension.

Together, the in vitro and in vivo data suggest a two-step model in which E-cadherin actively inhibits premature TJ formation in lower epidermal suprabasal layers while promoting TJ formation in the SG2 layer (Fig. 8g). In both cases, this may involve tension-dependent regulation of EGFR activity. In lower layers, all intercellular interfaces contain E-cadherin-AJs that are under low tension and permit moderate EGFR activity, thus further propagating low cortical tension and unstable TJs by promoting internalization. In contrast in the SG2 layer, AJs are absent from the apical intercellular interface, resulting in an anisotropy of actomyosin force distribution that may be sufficient to increase tension on E-cadherin junctions allowing for increased unfolding of α-catenin and recruitment of vinculin. This in turn reduces EGFR activity, thus further increasing cortical stiffness while at the same time stabilizing TJs at the cell surface. Whether TJ protein cell surface stability is directly controlled through phosphorylation by EGFR[55] or, indirectly, through regulation of actomyosin-dependent cortical tension[26, 56] is a question for future studies.

In conclusion, our data have a identified a mechanical circuit that integrates adhesion, cortical contractility, and biochemical signaling to drive the polarized organization of junctional tension necessary to build an in vivo epithelial barrier.

## Methods

**Generation of transgenic mice**. To generate E-cadherin$^{epi-/-}$ mice, female E-cadherin$^{flox/flox}$ mice were crossed to male E-cadherin$^{flox/wt}$;K14-Cre mice.

Transgenic E-cadherin$^{flox/flox}$ mice and K14-Cre mice and their genotyping were described previously[25, 57]. Lifeact-EGFP-expressing Ctr and E-cadherin$^{epi-/-}$ mice were generated by crossing E-cadherin$^{flox}$ mice with lifeact-GFP mice which were described previously[58]. All animals were on C57BL/6 N background.
Vinculin$^{epi-/-}$ mice were generated as follows. A genomic vinculin fragment isolated from a 129SvJ library was used to construct the vinculin targeting vector by standard cloning techniques. One loxP site was inserted 650 bp upstream of exon 22 and a second loxP site along with the neo cassette flanked by frt sites was inserted 1.8 kb downstream of the vinculin 3′UTR. Embryonic stem cells (129SvJ) were transfected with the linearized targeting construct by electroporation. 216 G418-resistant ES cell clones were screened for homologous recombination by southern blot analysis. For this, DNA was digested with BamHI, electrophoresed on a 0.8% agarose gel and blotted on nitrocellulose membrane. A 420 bp fragment corresponding to a region upstream of the homologous sequence of the targeting vector was used as southern blot probe. The fragment was generated by PCR using the primers 5′-AAGCCTTCCAACCTCAG-3′ and 5′-GAGAGAGAAAGG CAAGA-3′ and a vinculin clone from a genomic 129SvJ library as template. The probe was radiolabeled using rediprime labeling kit (GE healthcare) and (32 P) dCTP. DNA blots were hybridized and analyzed using a Phosphorimager (Fujifilm FLA-3000). Three individual homologous recombinant ES cell clones were used to generate chimeras by injecting them into blastocysts obtained from C57BL/6 females. Chimeras were backcrossed to C57BL/6 mice to achieve germ line transmission and with FLPase deleter mice to generate progeny with a floxed vinculin allele (VCL + /fl) without the neo cassette. Offspring carrying the floxed allele was identified by PCR using a primer pair which could distinguishing between the wild-type and the floxed allele (fw: 5′-CGG GTA AGA GCA CTG GCT GTT-3′ and rev: 5′-TCC ATA GGG CAG AGA TTT TTG3-′). Mice homozygous for floxed vinculin (VCL fl/fl) were crossed with transgenic mice expressing the cre recombinase under the control of the keratin14 promotor[25]. The recombination event was confirmed by PCR using primers surrounding the floxed region (rec-fw: 5′-CTT GAG TTG TCT GGG TGT GAG TAG A-3′ and rec-rev: 5′-AGA AAA TCA AGC AAA ACC T-3′). All mice were kept in a barrier facility. Animal care and experimental procedures were in accordance with the institutional and governmental guidelines.

**Preparation of epidermal whole mounts**. Backskin was prepared from newborn mice and subcutaneous fat was removed with curved tweezers. The epidermis was mechanically, carefully peeled off from the dermis with ultrafine curved tweezers thereby separating the basal from the suprabasal layers with residual patches of basal cells left on the epidermal sheet. During the whole procedure the basal side of the sheet was kept floating on PBS$^{2+}$ (PBS supplemented with 0.5 mM MgCl$_2$ and 0.1 mM CaCl$_2$). Subsequently the epidermal sheet was fixed floating on 4% PFA on ice for 10 min, washed on PBS for 5 min and permeabilized with 0.5% TritonX100/ PBS for 1 h at room temperature (RT). The permeabilized sheet was washed for 5 min on PBS and blocked with 10% FCS/PBS for 30 min/RT. For staining epidermal sheet were cut into ca. 5 × 5mm pieces. The SC of the epidermis is prone unspecific binding of ABs. Additionally it cannot be permeabilized to allow AB permeation. Thus, all following steps were performed incubating the sheet from the basal side leaving the SC side dry. Primary ABs were diluted either in AB diluent solution (Dako, S3022) or in ZO-1 hybridoma supernatant and incubated over night at 4 °C. Secondary ABs including DAPI and Phalloidin were incubated for 2 h/RT. After each AB incubation the sheet was rinsed 3× with PBS and washed 3× for 10 min. Finally the stained sheet was mounted in 50 µl Mowiol.

**Isolation and culture of primary keratinocytes**. Primary keratinocytes isolated from newborn mice were cultured in DMEM/HAM's F12 (FAD) medium with low Ca$^{2+}$ (50 µM) (Biochrom) supplemented with 10% FCS (chelated), penicillin (100 U ml$^{-1}$), streptomycin (100 µg ml$^{-1}$, Biochrom A2212), adenine (1.8 × 10$^{-4}$ M, SIGMA A3159), L-glutamine (2 mM, Biochrom K0282), hydrocortisone (0.5 µg ml$^{-1}$, Sigma H4001), EGF (10 ng ml$^{-1}$, Sigma E9644), cholera enterotoxin (10$^{-10}$ M, Sigma C-8052), insulin (5 µg ml$^{-1}$, Sigma I1882), and ascorbic acid (0.05 mg ml$^{-1}$, Sigma A4034). For keratinocyte isolation newborn mice were killed by decapitation and incubated in 50% Betaisodona/PBS for 30 min at 4 °C, 1 min PBS, 1 min 70% EtOH, 1 min PBS, and 1 min antibiotic/antimycotic solution. Tail and legs were removed and complete skin incubated in 2 ml Dispase (5 mg ml$^{-1}$)/FAD ("Keratinocyte culture" section) solution. After incubation over night at 4 °C, skin was transferred onto 500 µl FAD medium on a 6 cm dish and epidermis was separated from the dermis as a sheet. Epidermis was transferred dermal side down onto 500 µl of TrypLE (ThermoFisher Scientific) and incubated for 20 min at RT. Keratinocytes were washed out of the epidermal sheet using 3 ml of 10%FCS/PBS. After centrifugation keratinocytes were resuspended in FAD medium and seeded onto Collagen type-1 (0.04 mg ml$^{-1}$) (Biochrom, L7213) coated cell culture plates. Primary murine keratinocytes were kept at 32 °C and 5% CO$_2$. E-cadherin$^{-/-}$/P-cadherin$^{KD}$ cells were generated by lentiviral transduction of E-cadherin-deficient keratinocytes using C14 shRNA directed against P-cadherin[23].

To induce classical cadherin dependent junction formation, cells were switched to high Ca$^{2+}$ medium (1.5–1.8 mM). Cultured cells were regularly monitored for mycoplasma contamination and discarded in case of positive results.

**Antibodies and inhibitors**. Primary AB used in this study: mouse monoclonal against actin (WB 1:20000, MP Biomedicals #0869100, clone C4); rabbit polyclonal against the c-terminus of α-catenin (IF 1:2000, Sigma #c2081); rat monoclonal against the stretch induced α18 epitope of α-catenin[59] (IF 1:500, kind gift from Akira Nagafuchi); mouse monoclonal AB against the cytoplasmic domain of E-cadherin (IF 1:200, BD Transduction Laboratories #610182, clone number 36); rabbit polyclonal against the cytoplasmic domain of the EGFR (IF 1:200, WB 1:1000, Santa Cruz #sc-03); rabbit polyclonal against the cytoplasmic domain of the EGFR (WB 1:1000, Millipore #06-847); rabbit monoclonal against the cytoplasmic domain of the EGFR (WB 1:5000, IF 1:250, Abcam #ab52894); rabbit polyclonal against pTyr1068 of the EGFR (WB 1:1000, Invitrogen #44-788 G); mouse monoclonal against ERK2 (WB 1:2000, BD Transduction Laboratories #610103, clone 33); rabbit monoclonal against phosphorylated Thr202/Tyr204 of ERK1/2 (WB 1:2000, Cell Signaling #4370); mouse monoclonal against GAPDH (WB 1:10000, Ambion #AM4300); rabbit polyclonal against Histone H3 (WB 1:1000, Abcam #ab8895); rabbit polyclonal against Keratin6 (IF 1:500, Covance #PRB-169P); rabbit polyclonal against the c-terminus of Merlin (IF 1:100, Santa Cruz #sc332); mouse monoclonal against c-myc (IF 1:2000, Cell Signaling #2276); rabbit polyclonal against myosin light chain2 (WB 1:1000, Cell Signaling #3672); rabbit polyclonal against phosphorylated Thr18/Ser19 of myosin light chain2 (IF 1:100, WB 1:1000, Cell Signaling #3674); rabbit polyclonal against phosphorylated Ser20 (human) of myosin light chain (IF 1:400, Abcam #ab2480); mouse monoclonal against the c-terminus of occludin (IF 1:400, Invitrogen #33-1500); rat monoclonal against the extracellular domain of P-cadherin (hybridoma supernatant[60], kind gift from Masatoshi Takeichi); mouse monoclonal against Rac (WB 1:1000, Sigma #R2650, clone 23A8); mouse monoclonal against Vinculin (IF 1:250, WB 1:2000, Millipore #MAB3574, clone VIIF9 (7F9)); rat monoclonal against ZO-1 (hybridoma supernatant[61], clone R26.4 C). Phalloidin was used to stain F-actin (IF 1:500, Sigma #P1951, TRITC conjugated). Secondary ABs were species-specific ABs conjugated with either AlexaFluor 488, 594, or 647, used at a dilution of 1:500 for immunofluorescence (Molecular Probes, Life Technologies), or with horseradish peroxidase ABs used at 1:5000 for immunoblotting (Bio-Rad Laboratories). Inhibitors used in this study: ML7 myosin light chain kinase inhibitor, used in a range of 10–40 μM (Sigma #I2764); Blebbistatin myosin inhibitor, used in a range of 1–20 μM (Sigma #B0560); Y27632 ROCK inhibitor, used in a range of 0.5–10 μM (SIGMA #688000); gefitinib EGFR tyrosine kinase function inhibitor, used in a range of 0.1–100 μM (LC laboratories #G4408); GÖ6976 PKCα/β1 inhibitor, used in a range of 10–1000 nM (Calbiochem #365250).

**Transfection**. Keratinocytes were transfected at 80–100% confluency with Viromer®Red (lipocalyx) according to the manufacturer's protocol. In brief 1.5 μg DNA were diluted in 100 μl Buffer E, added to 1.25 μl Viromer®RED and incubated for 15 min at RT. 33 μl transfection mix were used per well (24-well plate).

**Plasmids**. P-cadherin expression vector: mouse P-cadherin cDNA was amplified from Plasmid DNA and ligated into the HindIII/BamHI digested pEGFP-N3 vector. RFP tagged E-cadherin full length (EcadFL) and truncated (EcadΔβ; aa1-812) expression vectors: mouse E-cadherin cDNA was amplified from plasmid DNA and fused with RFP by directional cloning into pCS2 + using HindIII/XbaI restriction sites. Myc tagged E-cadherin expression vector: mouse E-cadherin cDNA was amplified from plasmid DNA and inserted into pCS2 + using HindIII/BamHI restriction.

**Transepithelial resistance measurement**. A total of $5 \times 10^5$ keratinocytes were seeded on transwell filters (Corning (#3460), 0.4 μm pore size). Cells were allowed to settle and then switched to 1.8 mM high Ca$^{2+}$ medium. Formation of TER was measured over time using an automated cell monitoring system (cellZscope, nanoAnalytics).

**BrdU incorporation assay**. BrdU incorporation was measured by using the colorimetric Cell Proliferation ELISA, BrdU from Roche. 5000 Keratinocytes were plated in a 96-well format. 1 Day after plating, cells were pulse labeled with BrdU for 3 h. Measurements were taken according to the manufacturer's instructions.

**Biotinylation-internalization assay**. Cells were plated on 6 cm dishes and switched to high Ca$^{2+}$ medium for 48 h. DMSO or gefitinib (1 μM) treatment was started after 24 h and kept until lysis. Cells were kept at 4 °C on ice during the procedure. After 3× washings with PBS$^{++}$ (0.1 mM CaCl$_2$; 0.5 mM MgCl$^2$). Cells were incubated in 1.5 mg/ml EZ-Link$^{TM}$ Sulfo-NHS-SS-Biotin (Thermo #21331) in PBS$^{++}$ for 30 min agitating. For surface proteins, plates were rinsed once and then washed for 20 min with PBS$^{++}$/Glycin (100 mM). For internalized protein samples, plates were incubated with medium at 37 °C for 30 min. Internalization samples were washed 3× with stripping buffer 1 (100 mM sodium 2-mercaptoethanol sulfonic acid, 50 mM Tris-HCl pH8.6; 100 mM NaCl; 1 mM EDTA; 0.2% BSA) for 20 min. Subsequently internalization samples were washed 1× in stripping buffer 2 (120 mM iodacetamide) for 10 min. To control for efficient stripping of surface bound biotin after internalization, additional control samples were stripped immediately after biotin labeling without medium incubation. After treatment, cells were lysed in RIPA buffer by end-over-end rotation for 30 min and then centrifuged at maximum speed for 10 min in a benchtop centrifuge. A total of 400 μg

of protein were used to pulldown the biotinylated protein using 100 μl slurry of NeutrAvidin-Agarose beads (ThermoFisher #29202). Beads were equilibrated 2× in RIPA prior to pulldown. Lysates were incubated for 1 h at 4 °C with end-over-end rotation. Beads were washed 3× with 1 mL RIPA and centrifuged for 1 min at 1000 rpm. Eventually, proteins were eluted in 30 μl 2×Laemmli buffer, boiled for 10 min at 95 °C.

**Microscopy**. Confocal images were obtained with a Leica TCS SP8, equipped with a white light laser and gateable hybrid detectors. Objectives used with this microscope: PlanApo ×63, 1.4 NA; PlanApo ×40, 1.3 NA. Images to be used for deconvolution were obtained at optimal resolution according to Nyquist. Epi-fluorescence images were obtained with a Leica DMI6000. Objectives used with this microscope: PlanApo ×63, 1.4 NA; PlanApo ×20, 0.75 NA.

**Image processing and analysis**. Huygens deconvolution software (Scientific Volume Imaging, 15.10 release) was used to deconvolve confocal stacks (theoretical determination of the point spread function). Confocal stacks either deconvolve or not were processed with either Imaris (Bitplane, 7.0.0 release) to generate sub volume projections and virtual sections or Fiji[62]. Where required epifluorescence images were subjected to background subtraction (Fiji) using the rolling ball algorithm (radius 20 pixels).

**Quantification of α-catenin in different epidermal layers**. Stack profiles of fluorescence intensity of α-catenin and α18-epitope stainings of epidermal whole mounts were generated using the Leica LASX software and α18 distribution was normalized to total α-catenin distribution.

**Quantification of F-actin in different epidermal layers**. Orthogonal views of confocal stacks from epidermal whole mounts were generated using Fiji (5 views per biological replicate). The segmented line tool was used to measure mean fluorescence intensity of different layers (line width = 1 layer). The layer intensities of each view were normalized to the intensity of the respective spinous layer and the fold-increase of the SG2 and SG1 layer were plotted.

**Quantification of cortical F-actin, ZO-1 and EGFR enrichment in spinous and granular layer 2**. The Leica LASX software was used to analyze confocal plains of the respective epidermal layer in stacks of epidermal whole mounts. Lines of 10 μm length, 5 μm width were drawn across cell–cell contacts and the ratio of the mean fluorescence intensities of the ends (cytoplasmic, 3 μm/end) and the middle (junctional, 0.5 μm) of the line was formed. All ratios were normalized to values of the Ctr spinous layer, thereby displaying fold increase in Ctr SG2 and Ecad$^{epi-/-}$ spinous and SG2 layer.

**Quantification of junctional enrichment in different epidermal layers**. Segmented line tool (Fiji; line width 10 pixels) was used to measure mean fluorescence intensity for vinculin, ZO-1, EGFR at intercellular junctions of the basal, spinous, and granular layer 2. > 10 intercellular boarders were measured per layer and biological replicate. Number of biological replicates is annotated in the figure legends. To normalize for overall different fluorescence intensities between different experiments, the ratio of fluorescence intensities between layers of each experiment were compared.

**Quantification of adhesion zipper length**. The line tool (Fiji) was used to measure the length of primordial adhesion sites

**Quantification of Vinculin recruitment to intercellular adhesion sites**. The Leica LASX software was used to draw lines of 10 μm length along primordial adhesion sites from cytoplasm to cytoplasm of contacting cells. Fluorescence values along the line were normalized to the outermost cytoplasmic values.

**Traction force microscopy**. TFM measurements and calculations were performed as described previously[24]. In brief, substrates were generated by spin-coating silicone elastomer (CY52-276A and CY52-276B in 1:1 ratio, Dow Corning Toray) onto glass coverslips. The substrate contained two diffuse monolayers of fluorescent beads, one for reference (between the glass and elastomer) and the other just below the surface of the gel. The substrate was coated with 0.2 mg ml$^{-1}$ fibronectin from bovine plasma (Sigma-Aldrich) in PBS, which sat for 20 min at RT before being washed off with PBS. Ctr or E-cadherin$^{-/-}$ keratinocytes were then plated at low density in low-calcium medium. These cells were allowed to adhere and grow into pairs or small clusters before being switched to high-calcium medium for ~24 h before imaging the cells, the beads in their stressed positions, and the beads in their unstressed positions after removing cells by applying 0.5 mg ml$^{-1}$ proteinase K for 5 min. For measurements of intercellular tension, the vector sum of tractions under cell 1 was inferred to be the vector tension, $T_{21}$ exerted by cell 2 of the pair at the site of the intercellular adhesion, and vice versa for $T_{12}$[22, 63]. The magnitudes of these two tensions, $T_{21}$ and $T_{12}$, were divided by the spread area, $K_2$ or $K_1$, of cell 2 or cell 1, respectively, because traction scales directly with keratinocyte spread area[64]. $T_{21}/K_2$ and $T_{12}/K_1$ were then averaged for each pair to

generate the intercellular tension per cell area. For cell clusters, profiles of traction as a function of distance from colony edge and the average distance required to capture 75% of the strain energy exerted by the cell cluster were measured and calculated as described[24].

**Force indentation spectroscopy**. AFM indentation experiments were performed with a Bioscope II head (Veeco) mounted onto an Olympus IX73 microscope. For micromechanical measurements, spherical silicon dioxide beads with a diameter of 3.5 μm glued onto tip-less silicon nitride cantilevers (NanoAndMore, CP-PNPL-SiO-B-5) with a nominal spring constant of 0.08 N m$^{-1}$ were used. For all indentation experiments, forces of up to 3 nN were applied, and the velocities of cantilever approach and retraction were kept constant at 2 μm s$^{-1}$ ensuring detection of elastic properties only. All measurements were performed at 37 °C. Data analysis was carried out with AtomicJ using the Hertz model of impact[65].

Further detailed information about skin sample preparation, immunohistochemistry, and protein isolation and immunoblotting can be found in the Supplementary Methods.

**Statistics and repeatability of experiments**. The numbers of independent experiments and biological replicates performed for all experiments, $P$ values and the statistical tests that were used are indicated in the figure legends.

**Data availability**. The authors declare that the data supporting the findings of this study are available within the paper and its Supplementary Information files. Additional data are available from the corresponding author upon reasonable request.

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

## Acknowledgements

We would like to thank Christian Michels for help with barrier assays, members of the Niessen lab, Sandra Iden, and Hisham Bazzi (University of Cologne) and especially Kathleen J. Green (Northwestern University), Rudolf Merkel, and Bernd Hoffmann (Julich Research Center) for discussion and critical input. We also thank Akira Nagafuchi for the generous gift of the α18 AB and Franscesca Mascia and Stuart Yuspa (NIH) for providing us with tissue sections of epidermal EGFR knockout mice. Furthermore we greatly acknowledge Sabine Eming (University of Cologne) for providing human skin samples. We greatly acknowledge the funding by DFG: SFB 829 A1, Z2, and SPP 1782 grant no. NI1234/6-1.

## Author contributions

M.R. and C.M.N. conceived of the study. M.R., A.F.M., A.K., S.A.W., M.A. and C.M.N. designed experiments. M.R., A.F.M. and G.G.B. carried out experiments and analyzed data. S.M., M.M. and W.Z. generated transgenic vinculin-floxed mice, C.J. and A.S. assisted with imaging and image analysis. M.R. and C.M.N. wrote the manuscript. All authors provided intellectual input, vetted, and approved the final manuscript.

## Additional information

**Competing interests:** The authors declare no competing financial interests.

