## [Peer Review File · Nature Communications]

Reviewers' comments:

Reviewer #1 (Remarks to the Author):

This is a very nice, well written paper documenting the coordination between E-cadherin, tight junction proteins, tension, and epidermal growth factor receptor in regulating apical basolateral polarity of cells in the epidermis. The main findings are based on whole mount images of stratified epidermis and the use of tissue specific genetic ablation. Their use of whole tissue mounts to address how junctions organize between cells in complex tissues is a major advance over the majority of similar studies that relied solely on cultured epithelial cell lines. The quality of the data are high, and the data analyses and use of statistics are appropriate.

There are several interesting findings reported here. However, the abstract states that their results 'reveal a mechanistic role for EGFR at TJs and uncover why EGFR inhibitors compromise skin barrier function', but in actuality, the data presented do not go that far. There are several correlations, but some significant gaps in mechanism weaken some of their arguments. The following are considerations and suggestions for studies that would make a more compelling evidence for their claims.

1. The role of tension on E-cadherin in regulating this polarized organization is not definitively established. The reported correlations are compelling, but recent findings indicated that ZO-1 controls cell-cell tension at endothelial adherens junctions (Tornavaca et al 2015 J Cell Bio). Given that these authors did not knock down any tight junction proteins, it is difficult to rule out a reciprocal role of tight junction proteins in tension regulation.
2. On page 8, they address the role of P-cadherin, and ask whether its upregulation in E-cadherin -/- keratinocytes might compensate for E-cadherin loss. They demonstrate both P- and E-cadherin support tension, based on alpha catenin epitope exposure and vinculin enrichment, but P-cadherin did not compensate for E-cadherin. They claim that this indicates that zippering requires a threshold level of cadherins, but don't actually prove this by overexpressing P-cadherin. They should also consider recent results by Bazellieres et al 2015 Nat Cell Biol that indicated different mechanical roles of P- and E-cadherin.
3. The traction force result showing that loss of E-cadherin affected tension between cultured cells affects tension is not particularly surprising. However, to relate this more closely to the intact tissue, were the control cells cultured for sufficient times to develop tight junctions, or to establish apico-basolateral polarity? In such cases, how would ZO-1 knock down, for example, affect the results (see comment #1 regarding ZO-1)? Also, how does P-cadherin expression affect the traction data (see #2 above).
4. The vinculin results nicely show that vinculin is not essential for tissue maturation in vivo, but appears to be most important in early formation of focal adherens junctions. Vinculin is itself not a force sensor (the authors should correct this), but its enrichment at AJs under tension appears to require specific phosphorylation (see Bays et al J Cell Biol 2014). In cultured cells, vinculin phosphorylation is transient, suggesting that vinculin enrichment would decrease at mature AJs. How do they reconcile this result (page 9) with vinculin enrichment in AJs under tension in SG2? Are there differences in kinase signaling in these contexts that might explain the difference?
5. The E-cadherin dependence of EGFR localization is intriguing, especially in the context of cell

polarity. However, it is unclear from the data provided how E-cadherin integrates EGFR signaling and mechanotransduction as the title states. The data currently presented don't convincingly address how EGFR signaling and localization contributes to polarization and permeability. The causal relationships between E-cadherin adhesion and signaling, EGFR localization and signaling, tension increases, and ZO-1 localization are currently unclear. The correlation with EGFR localization is also intriguing in light of studies by Collins et al 2012 *Curr Biol* and more recently by Muhamed et al 2016 *J Cell Sci* that linked force transduction to growth factor-dependent changes in cell contractility. Given that increased EGFR activity in E-cadherin depleted cells doesn't influence proliferation, this raises questions as to what is activating EGFR and how signals could be affecting AJ organization and tension distributions.

Reviewer #2 (Remarks to the Author):

Establishment of polarized epithelial barrier is a critical process that underlies important function of essentially all epithelial organs. The architecture and organization of specialized cell-cell junctional complexes, adherens junctions (AJs) and tight junctions (TJs), have been studied in detail in simple epithelium but much less is known about how these junctions are organized in multilayered epithelium. Here, Rüksam et al. have addressed this issue by combining genetic models and high-resolution imaging of epidermal whole-mounts with additional cellular model systems including measurements of junctional traction forces. Tight junctions have long been known to be restricted to the uppermost layers of SG in the epidermis (particularly by using occluding staining; eg. Tunggal et al (2005) but also Morita K, Itoh M, Saitou M, Ando-Akatsuka Y, Furuse M, Yoneda K, Imamura S, Fujimoto K, Tsukita S (1998) Subcellular distribution of tight junction-associated proteins (occludin, ZO-1, ZO-2) in rodent skin. *J Invest Dermatol* 110: 862-866). However, the mechanical properties of the individual cell layers have not been addressed in detail.

The authors report that while AJs are found throughout the epidermal layers ZO-1-enriched TJs form only at the uppermost cell layer in stratum granulosum (SG). Using immunocytochemistry of selected marker proteins the authors conclude that the TJ-containing SG layer differs mechanically from the more basal cell layers and is under high mechanical tension as evidenced by AJ recruitment of vinculin. Deletion of vinculin disrupted the AJ/TJ organization *in vitro* but surprisingly did not significantly affect junctional tissue polarity in the epidermis. A model is proposed wherein the restricted polarized tissue localization of tight junctions is achieved through E-cadherin-mediated modulation of EGFR-activation specifically at the uppermost cell layer. EGFR-dependent signaling is then somehow linked to the formation of TJs and elevated tension at the uppermost SG layers. To this end it is shown that EGF-treatment inhibits TER in normal but not in E-Cad null keratinocytes.

This is an interesting study in which the experimental data presented appears to be of high quality throughout the manuscript. Statistical analyses have been performed with seemingly appropriate tests (having said that, this reviewer is not a statistical expert). The manuscript is well-written although the description of the proposed model is a bit difficult to grasp. Phenotypic observations are rather solid and likely very interesting for researchers from several fields. However, very little mechanistic insight accompanies these findings and some additional clarifications are necessary to fill in some of the gaps to strengthen the proposed model.

Major comments:

- 1) Evidence for the higher tension at the uppermost SG layer when compared with the underlying layers is quite convincing (in sections) but the experimental setup where the link between E-Cad and junctional tension is examined (Fig. 5g, h) is poorly described. Reference for traction force

microscopy has been given in wrong format. Number is given but other references are shown with names as is the reference list. Which reference the "11" is remains unclear - Mertz 2013? The experimental details need to be better explained. How were the substrates used in these coated? With FN or E-Cad-fragments? If it was the former it needs to be addressed whether E-Cad (but not P-Cad) synergizes with integrin-mediated adhesions to regulate global contractility, possibly via regulation of EGFR-signaling as described recently (Muhammed et al. 2016). If such synergy drives contractility where would integrins be anchored in vivo in SG2? Or is that the reason why a difference is observed in vivo vs. in vitro? In order to focus on the lateral contractility spanning the adherens an alternatively approach could be to measure the tension at adherens junctions by laser ablation (eg. as in Huvencuers et al 2012)).

2) What accounts for increased vinculin staining at uppermost layers? Vinculin is recruited at junctions upon mechanical tension but is it also generally stabilized? Or is it under transcriptional control? Otherwise one would not expect an expression intensity "gradient", only a difference in localization (AJs vs. cytoplasm).

3) The discrepancy between the ZO-1 localization and cell-cell adhesion patterns in control vs. vinculin null keratinocytes in vitro vs. in vivo (ZO-1 pattern) is confusing. A PKC-dependent translocation mechanism of ZO-1 from TJs to focal adhesions has been described. Are the ZO-1-positive patterns in vinculin-/- keratinocytes integrin-dependent? Do they co-localize with integrins instead of E-Cad in vinculin null keratinocytes?

4) What about the P-cadherin/E-cadherin zipper-patterns in in vitro keratinocytes? Are those perhaps integrin-dependent? Are they differentially sensitive to changes in substrate stiffness (might be linked to the possibility that SG2 layer could be in contact with more rigid Stratum Corneum).

5) The role of EGFR: The staining for EGFR in fig 7. is of suboptimal quality. It is not clear if EGFR in the uppermost SG layer is intracellular (endocytosed => active EGFR-signaling?) or apical (mislocalized)? Which cells produce the most EGF or do they all produce it?

6) What happens to junctional components in in vitro cultured normal and E-Cad-null keratinocytes upon addition of EGF? Are they endocytosed together with EGFR?

Minor comments:

- An open question is where the force comes in the uppermost layer? E-Cad is found in all layers but why only uppermost layer forms contractile TJs? Could mechanical properties of the overlaying Stratum Corneum play a role (SC has been suggested to be stiffer than the viable layer perhaps due to loss of water that might cause the SC layer to "shrink")? Or is it anticipated that the tension is due to "stretching" of less proliferating squamous top cell layers by the pressure from basal layers proliferating and pushing upwards? These issues should be shortly discussed in order to better describe how the findings in this study come together to support the proposed model.

- The underlying logic is difficult to follow in discussion of the model that incorporates the observations that both E-Cad mediated stimulation or strong inhibition of EGFR compromise TJ formation into the findings in E-Cad null epidermis. Especially since more modest inhibition of EGFR in E-Cad-/- keratinocytes instead facilitated TJ-formation? How do these come together? Moreover, in simple epithelium EGF appears to increase TER (for example the reference "Epidermal growth factor receptor activation differentially regulates claudin expression and enhances transepithelial resistance in Madin-Darby canine kidney cells." Singh AB et al. J Biol Chem. (2004) and others). Why should this be the opposite in keratinocytes? This could maybe be rephrased in the discussion to clarify the links between the observations.

- Not clear whether and how much EGFR is enriched at SG2 - compare fig7 and Fig 8a!
- On page 9 line 281 the reference to (Fig. 6f,g) now says 5f,g
- References have some issues with symbols (at least Bueler 2013, Shen 2005 and Yonemura 2011)

Reviewer #3 (Remarks to the Author):

This paper addresses the question whether E-cadherin-mediated force transduction in the epidermis is important for epithelial differentiation and tight junction formation. Using a knockout approach, E-cadherin deletion is shown to lead to reorganization of the actin cytoskeleton and components known to be involved in force transduction such as vinculin. As E-cadherin is known to lead to a defect in tight junction formation in the skin and they find that force transmission is affected in cultured keratinocytes, the authors postulate that mechanical force controls the formation and positioning of tight junctions in the epidermis. They further conclude that the tissue distribution of EGFR is also deregulated and that E-cadherin may regulate junction formation by controlling the timing of EGFR signaling.

Adherens junctions are major transducers of mechanical force between neighboring cells. E-cadherin is in many epithelia the main cell-cell adhesion protein of adherens junctions and is a component of the force-transducing molecular linker. Interplay between EGFR signaling, E-cadherin and intercellular junctions has been analyzed in different systems and is known to be a key mechanism of epithelial differentiation and junction formation. However, little is known about these mechanisms *in vivo* and, in particular, the role they play in the differentiation and function of stratified epithelia. The paper thus addresses an important and timely question.

The paper presents a considerable amount of high quality data that point to potentially exciting conclusions. However, much of the presented evidence is indirect and incomplete. Moreover, sometimes it is difficult to reconcile the actual figures with the description in the text. Hence, the main conclusions are not well supported.

1) A major drawback of the experimental approach is the lack of experiments directly measuring tension changes in the epidermis. While the staining of specific alpha-catenin epitopes and vinculin may provide indications of tension alterations, they do not provide a direct readout for tension. Admittedly, some methods used to measure tension may not work *in vivo* and in stratified epithelia (e.g., ablation experiments) and others are cumbersome to implement (tension sensors). However, to substitute such measurements with an *in vitro* method that makes use of isolated pairs of cells to mention tension is inappropriate. In fact, some of the results shown here quite clearly indicate that the *in vitro* and *in vivo* systems do not yield the same results. Instead of traction force microscopy, which is difficult to implement with monolayers, one would expect at least *in vitro* force measurement using continuous layers of cells (e.g., laser ablation or tension sensors).

2) As the vinculin deletion results do not support the model of tension and tight junction positioning, it is surprising that these experiments were not completed by measuring the effect of vinculin depletion on cell-cell tension using the cultured keratinocytes. Based on the data as they are, one could also speculate that the vinculin deletion results mean that tissue tension is irrelevant for tight junction organization and epidermal differentiation.

3) It is unfortunate that only ZO-1 is used as a marker for tight junctions, as it is clearly not specific for that junction in the epidermis. It is often not clear why some areas positive for ZO-1

are marked as being tight junctions and others are not. For example, in Figure 4e, the image at E16.5 shows staining of all cell layers by ZO-1. Some areas are declared as tight junctions but without any further support for such a conclusion. The image clearly illustrates that it is not sufficient to stain for ZO-1 to identify tight junctions. More specific markers should be added to the analysis, preferably membrane proteins such as occludin.

4) Fig. 1b - In contrast to the description in the text (page 4), there seems to be a considerable amount of vinculin staining in layers lower than SG2. Where one can see E-cadherin, there is vinculin. Similarly in Fig. 3, phalloidin in spinous layer: It seems an overstatement to describe the right panel as a 'strongly enhanced cortical' staining. It is hardly visible. Moreover, the vertical section through the layers underneath seems to suggest that the phalloidin staining is generally stronger also in the SG layers. Here too, staining a more specific tight junction marker would considerably strengthen the conclusion that E-cadherin deletion leads to redistribution of tight junction proteins. Panel e seems to suggest a surprisingly small effect on ZO-1 and whether that alone would be sufficient to lead to a loss of function is questionable.

5) Page 5, lines 138/9 - While there is clearly vinculin staining in the proximity of tight junctions, these images do not offer the resolution to demonstrate that this is tight junction associated vinculin. Figure 1 shows a similar distribution of E-cadherin and a complete overlap with vinculin in SG2. How is that be compatible with vinculin in tight junctions?

6) Page 8, figures 4 and supplementary 5 - The increase in Rac activity is not convincing as there seems to be an increase in total Rac as well. One wonders why only Rac is investigated as RhoA may directly lead to tension.

7) Why did the authors choose to investigate only ppMLC2 phosphorylation and not also single phosphorylation (the text confuses these two different forms). Based on supplementary figure 5f, there does not seem to be any active myosin at the junction/membrane even in controls, which seems surprising. How do the authors envision that cortical tension is generated? How is the distribution of ppMLC and pMLC in vivo?

8) page 8: the authors seem to ask whether abolition of tension leads to tight junction reformation by inhibiting MLCK. This experiment would be far more conclusive if myosin were inhibited directly with blebbistatin. Why was only MLCK studied and not the RhoA/ROCK pathway?

Minor comments:

1) Fig. 1c - The alpha18 antibody staining should be shown as an individual channel so that the gradient in staining becomes more evident.

2) Fig. 1g - The term 'subapical' is confusing. It seems that the authors are referring to lateral junctions. This term normally refers to an epithelial region between tight junctions and the apical domain.

3) Page 5 - The conclusions of the first results section (lines 115-126) should be phrased more carefully. The experiments reported used indirect markers for cell-cell tension and no actual tension experiments are shown. The data are supportive of the described tension gradients but do not prove them.

4) Page 6, line 180 - This statement reads as though loss of E-cadherin would lead to formation of functional tight junctions elsewhere in the skin; however, I don't assume this is what the authors intended to conclude. The data just suggest a minor redistribution of ZO-1 compatible with the previously reported loss of functional tight junctions in such animals.

5) Figure 4/page 7 - The section makes a statement about functional barrier formation. Where is it shown whether the barrier is functional or not? Do the authors rely on previous publications?

6) Fig. 8a/b - this is really a central figure supporting the main conclusion and should hence be quantified.

7) Fig. 8c - Why are the Ecad^{-/-} cells incubated with DMSO and not like the controls?

8) The discussion concludes that a mechanical circuit was identified that integrates adhesion, cortical contractility and chemical signaling etc. The paper does not contain any evidence for tension regulating EGFR signaling or for altered cortical contractility (which would require altered myosin activity at the cortex).

We would like to thank all three reviewers for their constructive suggestions, comments and ideas, which we feel has strongly improved the data and impact of our manuscript.

Reviewer #1

This is a very nice, well written paper documenting the coordination between E-cadherin, tight junction proteins, tension, and epidermal growth factor receptor in regulating apical basolateral polarity of cells in the epidermis. The main findings are based on whole mount images of stratified epidermis and the use of tissue specific genetic ablation. Their use of whole tissue mounts to address how junctions organize between cells in complex tissues is a major advance over the majority of similar studies that relied solely on cultured epithelial cell lines. The quality of the data are high, and the data analyses and use of statistics are appropriate.

There are several interesting findings reported here. However, the abstract states that their results 'reveal a mechanistic role for EGFR at TJs and uncover why EGFR inhibitors compromise skin barrier function', but in actuality, the data presented do not go that far. There are several correlations, but some significant gaps in mechanism weaken some of their arguments. The following are considerations and suggestions for studies that would make a more compelling evidence for their claims.

We would like to thank the reviewer for appreciating the importance of our study and the quality of the data and for the very helpful comments. Based on the reviewers suggestion we have now added significant amounts of new data that provide more mechanistic insight into how E-cadherin controls tension and barrier formation. We find that E-cadherin, through the EGFR, controls not only the formation of functional tight junctions but also actomyosin-mediated cortical stiffness of a multi-layered barrier forming epithelium. In addition, myosin activity is required for tight junction formation. Together these new data thus provide further links between E-cadherin control of tissue polarity, tension and tight junctions. We also show that actomyosin-dependent regulation of tension controls EGFR activity, thus indicating tight positive feedback control between mechanical and chemical signaling in keratinocytes. Finally, we provide evidence that E-cadherin, through the EGFR, controls cell surface stability of the tight junctional protein occludin, providing a molecular mechanism for the dysfunctional TJ barrier upon loss of E-cadherin or over-activation of EGFR. The specific comments are addressed in detail below.

1. The role of tension on E-cadherin in regulating this polarized organization is not definitively established. The reported correlations are compelling, but recent findings indicated that ZO-1 controls cell-cell tension at endothelial adherens junctions (Tornavaca et al 2015 J Cell Bio). Given that these authors did not knock down any tight junction proteins, it is difficult to rule out a reciprocal role of tight junction proteins in tension regulation.

We have now added new experiments that strengthen the conclusion that E-cadherin controls tension and that tension is important for restricted formation of TJ barriers in the epidermis. We established an *in vitro* system, where primary keratinocytes, upon switching to high Ca^{2+} (1.8 mM), form a confluent multilayered sheet in which tight junctions specifically form at the most apical surface (Vaezi et al., Dev. Cell, 2002; Michels et al., NY Acad. Sci, 2009). We then used atomic force microscopy (AFM)-based force indentation spectroscopy to quantify actomyosin-driven cortical stiffness as a readout for the arrangement of the actomyosin cortex and cortical tension. These experiments demonstrated that E-cadherin controls cortical stiffness of the most upper apical layer where the TJs are formed (new Fig. 5g-i). Furthermore, inhibition with blebbistatin lowered cortical stiffness, directly demonstrating that this

stiffness depends on actomyosin activity (new Supplementary Fig. 5i). To rule out that E-cadherin controls tension indirectly through TJs we also depleted ZO-1, the cytoskeletal protein that links TJs to the actin cytoskeleton, in primary keratinocytes, and found that this did not affect cortical stiffness as measured by AFM, indicating that reduced levels of junctional ZO-1 cannot explain alterations in reduced stiffness upon E-cadherin loss (new Supplementary Fig. 5j).

2. On page 8, they address the role of P-cadherin, and ask whether its upregulation in E-cadherin -/- keratinocytes might compensate for E-cadherin loss. They demonstrate both P- and E-cadherin support tension, based on alpha catenin epitope exposure and vinculin enrichment, but P-cadherin did not compensate for E-cadherin. They claim that this indicates that zippering requires a threshold level of cadherins, but don't actually prove this by overexpressing P-cadherin. They should also consider recent results by Bazellieres et al 2015 Nat Cell Biol that indicated different mechanical roles of P- and E-cadherin.

[Redacted]

3. The traction force result showing that loss of E-cadherin affected tension between cultured cells affects tension is not particularly surprising. However, to relate this more closely to the intact tissue, were the control cells cultured for sufficient times to develop tight junctions, or to establish apico-basolateral polarity? In such cases, how would ZO-1 knock down, for example, affect the results (see comment #1 regarding ZO-1)? Also, how does P-cadherin expression affect the traction data (see #2 above).

Although at first sight our result on alterations in junctional tension upon loss of E-cadherin seems not very surprising, it is perhaps more so when taking into account that these primary keratinocytes were kept for 24 hours in high calcium, allowing also desmosomes to be fully assembled in these cells (Michels et al., J. Invest. Dermatol., 2009). These junctions are considered to provide key mechanical strength to primary keratinocytes, as mutations or antibodies to desmosomal cadherins cause skin blistering and loss of cell-cell contacts. In addition, recent data (Ramms et al., Proc. Natl. Acad. Sci. U. S. A., 2013; Kröger et al., J. Cell Biol., 2013) indicate that keratins coupled to junctions such as desmosomes provide

crucial tensile force in keratinocytes. We thus considered it absolutely essential to show that E-cadherin directly controls intercellular tension.

Based on the comments of this and the other reviewers we have made significant efforts to quantify tension in the tissue context or in a context of a TJ-forming multilayered epithelium. Following the suggestion of Reviewer 3, we worked hard to establish in vivo junctional laser ablation to quantify junctional tension in newborn mice and E16.5 embryos when junctional tissue polarization is first observed. However, these experiments were technically extremely challenging as junctional tissue polarization coincides with stratum corneum barrier formation and this dead cornified layer is a major obstacle for laser penetration and energy dissipation at this point. To overcome this stratum corneum barrier, we established an in vitro system where extended Ca^{2+} treatment of confluent primary keratinocytes leads to multi-layering and specific formation of tight junctions at the most apical layer, resembling the 3D complexity of the in vivo system (Vaezi et al., Dev Cell 2002). However, laser ablation in this system, unlike in simple monolayers where it is commonly used, revealed a very complex pattern of tension regulation and dissipation that is likely due to the three dimensional nature of the system, precluding the quantification of 2D vertex displacement as a readout for junction tension. We therefore used AFM-based force indentation spectroscopy to quantify apical cortical stiffness, which we as well as others show to directly depend on the tension generated by cortical actomyosin network (new Supplementary Fig. 5i)(Lecuit & Lenne, Nat. Rev. Mol. Cell Biol., 2007; Salbreux et al., Trends Cell Biol., 2012). We observed a reduction in cortical stiffness upon loss of E-cadherin (new Fig. 5g-i) whereas depletion of ZO-1 did not have an effect (new Supplementary Figure 5j). We did not assess the role of P-cadherin alone in AFM under these conditions, as loss of P-cadherin alone does not alter tight junctional (TJ) skin barrier properties in vivo (Radice et al., JCB 1997; Tinkle et al., Proc. Natl. Acad. Sci. U. S. A., 2008) and in vitro (own unpublished results).

4. The vinculin results nicely show that vinculin is not essential for tissue maturation in vivo, but appears to be most important in early formation of focal adherens junctions. Vinculin is itself not a force sensor (the authors should correct this), but its enrichment at AJs under tension appears to require specific phosphorylation (see Bays et al J Cell Biol 2014). In cultured cells, vinculin phosphorylation is transient, suggesting that vinculin enrichment would decrease at mature AJs. How do they reconcile this result (page 9) with vinculin enrichment in AJs under tension in SG2? Are there differences in kinase signaling in these contexts that might explain the difference?

Vinculin has been studied extensively in focal adhesions and focal adherens junctions in simple epithelial and endothelial cell culture and together these studies showed that vinculin is recruited to both cadherin and integrin complexes in response to external force (see e.g. Grashoff et al., Nature, 2010; Yonemura et al., Nat. Cell Biol., 2010; Galbraith et al., J. Cell Biol., 2002), as also stated in the introduction of the (Bays et al., J. Cell Biol., 2014) paper. For these reasons we termed vinculin a force sensor, but after the comment of the reviewer have now refrained from using this term. We cannot exclude that there are differences in Abl and related kinase signaling in the SG2 versus lower layers that may explain the strongly enriched recruitment of vinculin in the SG2. How this relates to phosphorylation of the vinculin Y822 site in vinculin is an interesting topic in itself. However, as we found that vinculin was not essential for cortical tension as measured by AFM (new Fig. 6h and Supplementary Figure 6g,h), tissue polarization and TJs, more detailed investigation of this signaling is beyond the scope of this manuscript.

5. The E-cadherin dependence of EGFR localization is intriguing, especially in the context of cell polarity. However, it is unclear from the data provided how E-cadherin integrates EGFR signaling and mechanotransduction as the title states. The data currently presented don't convincingly address how EGFR signaling and localization contributes to polarization and permeability. The causal relationships between E-cadherin adhesion and signaling, EGFR localization and signaling, tension increases, and ZO-1

localization are currently unclear. The correlation with EGFR localization is also intriguing in light of studies by Collins et al 2012 Curr Biol and more recently by Muhamed et al 2016 J Cell Sci that linked force transduction to growth factor-dependent changes in cell contractility. Given that increased EGFR activity in E-cadherin depleted cells doesn't influence proliferation, this raises questions as to what is activating EGFR and how signals could be affecting AJ organization and tension distributions. (Collins et al show that localized tensional forces on PECAM-1 result in, global signaling responses. Force-dependent activation of PI3K promotes cell-wide activation of integrins and the small GTPase RhoA. These signaling events facilitate changes in cytoskeletal architecture, including growth of focal adhesions and adaptive cytoskeletal stiffening.)

We have now performed substantial new experiments that shed additional light on what activates EGFR and how these signals then affect tension distribution and AJ organization. First, we examined whether actomyosin driven tension impacts the activity of the EGFR and find that when lowering actomyosin tension without completely abolishing it, using a low dose of the myosin II inhibitor blebbistatin (10 μ M), this results in increased EGFR activity (new Fig. 8g). This same blebbistatin concentration also prevents the formation of functional TJs (new Fig. 5k), thus linking tension-mediated regulation of EGFR activity and TJ formation. On the other hand, we observe that inhibition of EGFR after junction formation is initiated, not only rescues TJ formation in E-cadherin^{-/-} primary keratinocytes (Fig. 8e), but also rescues the reduced cortical stiffness observed in these cells, thus directly linking rescue of TJ barrier formation with increased cortical tension (new Fig. 8f).

To directly assess whether EGFR activity controls TJs, we analyzed the effect of E-cadherin and EGFR on the cell surface stability of TJ proteins. To this end, we performed biotinylation-based internalization experiments and found increased internalization of occludin and EGFR but not P-cadherin or claudin-1 upon loss of E-cadherin. Importantly, these defects could be rescued by inhibition of EGFR activity (new Fig. 8d).

Our data thus indicate that E-cadherin-dependent tension regulation is required to spatiotemporally constrain EGFR activity in the SG2 to control organization and thus tension of the actomyosin cortex to stabilize and facilitate maturation of TJs. Together, the in vitro and in vivo data suggest a two-step model in which E-cadherin actively inhibits premature TJ formation in lower epidermal suprabasal layers while promoting TJ formation in the SG2 layer (new Fig. 8h). In lower layers, all intercellular interfaces contain E-cadherin AJs that are under low tension, which promotes moderate EGFR activity, preventing further buildup of cortical tension and destabilizing TJs by promoting internalization. In contrast in the SG2 layer, AJs are absent from the apical intercellular interface, resulting in an anisotropy of actomyosin force distribution that may be sufficient to increase tension on junctions, allowing for increased unfolding of α -catenin and recruitment of vinculin. These high-tension state AJs recruit EGFR and constrain its activity thus enhancing cortical stiffness and stabilizing TJs at the cell surface.

Reviewer #2

Establishment of polarized epithelial barrier is a critical process that underlies important function of essentially all epithelial organs. The architecture and organization of specialized cell-cell junctional complexes, adherens junctions (AJs) and tight junctions (TJs), have been studied in detail in simple epithelium but much less is known about how these junctions are organized in multilayered epithelium. Here, Rüksam et al. have addressed this issue by combining genetic models and high-resolution imaging of epidermal whole-mounts with additional cellular model systems including measurements of junctional traction forces. Tight junctions have long been known to be restricted to the uppermost layers of SG in

the epidermis (particularly by using occluding staining; eg. Tunggal et al (2005) but also Morita K, Itoh M, Saitou M, Ando-Akatsuka Y, Furuse M, Yoneda K, Imamura S, Fujimoto K, Tsukita S (1998) Subcellular distribution of tight junction-associated proteins (occludin, ZO-1, ZO-2) in rodent skin. J Invest Dermatol 110: 862-866). However, the mechanical properties of the individual cell layers have not been addressed in detail.

The authors report that while AJs are found throughout the epidermal layers ZO-1-enriched TJs form only at the uppermost cell layer in stratum granulosum (SG). Using immunocytochemistry of selected marker proteins the authors conclude that the TJ-containing SG layer differs mechanically from the more basal cell layers and is under high mechanical tension as evidenced by AJ recruitment of vinculin. Deletion of vinculin disrupted the AJ/TJ organization in vitro but surprisingly did not significantly affect junctional tissue polarity in the epidermis. A model is proposed wherein the restricted polarized tissue localization of tight junctions is achieved through E-cadherin-mediated modulation of EGFR-activation specifically at the uppermost cell layer. EGFR-dependent signaling is then somehow linked to the formation of TJs and elevated tension at the uppermost SG layers. To this end it is shown that EGF-treatment inhibits TER in normal but not in E-Cad null keratinocytes.

This is an interesting study in which the experimental data presented appears to be of high quality throughout the manuscript. Statistical analyses have been performed with seemingly appropriate tests (having said that, this reviewer is not a statistical expert). The manuscript is well-written although the description of the proposed model is a bit difficult to grasp. Phenotypic observations are rather solid and likely very interesting for researchers from several fields. However, very little mechanistic insight accompanies these findings and some additional clarifications are necessary to fill in some of the gaps to strengthen the proposed model.

We would like to thank the reviewer for appreciating the quality of our data and its importance for several fields. We have now added several experiments that we believe substantially strengthen mechanistic insights into how E-cadherin integrates mechanical and EGFR receptor signaling and how this regulates the restricted formation of TJs to the granular layer. We furthermore integrated these new data to further refine and better describe the proposed model. We discuss these new data in detail below.

Major comments:

1) Evidence for the higher tension at the uppermost SG layer when compared with the underlying layers is quite convincing (in sections) but the experimental setup where the link between E-Cad and junctional tension is examined (Fig. 5g, h) is poorly described. Reference for traction force microscopy has been given in wrong format. Number is given but other references are shown with names as is the reference list. Which reference the "11" is remains unclear - Mertz 2013? The experimental details need to be better explained. How were the substrates used in these coated? With FN or E-Cad-fragments? If it was the former it needs to be addressed whether ECad (but not PCad) synergizes with integrin-mediated adhesions to regulate global contractility, possibly via regulation of EGFR-signaling as described recently (Muhamed et al. 2016). If such synergy drives contractility where would integrins be anchored in vivo in SG2? Or is that the reason why a difference is observed in vivo vs. in vitro? In order to focus on the lateral contractility spanning the adherens an alternatively approach could be to measure the tension at adherens junctions by laser ablation (eg. as in Huveneers et al 2012).

We apologize for the poor description of the traction force microscopy (TFM) and for the unformatted reference for TFM and have now improved and corrected this. The technical details of the TFM, including substrate preparation, were described in our Mertz et al. PNAS 110:842-7, 2013 paper. In brief, a

monolayer of fluorescent beads was embedded in a thin silicone gel coated with fibronectin before cells were seeded, as described in the paper above.

Importantly, we applied TFM on pairs of intercellular adhesive cells as this allows us to assess how loss of E-cadherin affects magnitudes of intercellular tension. In this case tension on intercellular junctions is directly coordinated with cell-matrix junctions see e.g. (Lui et al, Proc. Natl. Acad. Sci. U. S. A. 2010; Maruthamuthu et al., Proc. Natl. Acad. Sci. U. S. A., 2011). Our data show that E-cadherin in primary keratinocytes contributes to intercellular tension, despite the presence of desmosomes that are considered to be the major mechanical adhesive system in keratinocytes. The traction force data on keratinocyte colonies show that E-cadherin contributes to the coordination of mechanical responses under conditions in which in cells have not formed stratified, multi-layered epithelium yet. As published in the 2013 Mertz PNAS paper, we have already shown that upon additional loss of P-cadherin these colonies interact mechanically with their substratum as single mechanical units, indicating that both P-cadherin and E-cadherin coordinate global contractility in cooperation with integrin-ECM interactions (Mertz et al., Proc. Natl. Acad. Sci. U. S. A., 2013).

Nevertheless, as also reviewers 1 and 3 point out, these conditions do not fully recapitulate conditions that represent the epidermal tissue, in which the highest tension is seen in cells that are not in direct contact with extracellular matrix and thus do not have adhesive integrin-ECM contacts.

As suggested by this reviewer and reviewer 3 we tried to set up laser ablation experiments in *in vivo* mammalian newborn and E16.5 back skin epidermis to determine junctional tension between SG2 cells. However, the physical properties of the stratum corneum, a very stiff protein-lipid cross-linked cellular network, made it impossible to reach reproducible and efficient ablation in the SG2 layer. Despite using a range of conditions as well as different lasers (UV or far-red 2-photon) and microscope systems, ablation of junctions was extremely difficult to achieve *in vivo*. This *in vivo* approach is further complicated by the fact that keratinocytes in the epidermis are tightly interconnected by intercellular junctions not only at the lateral but also apical and basal surface. Thus, a recoil of tricellular junctions that is used as a readout in simple epithelia *in vitro* or in a free floating epithelium in e.g. *Drosophila* *in vivo* is technically difficult to achieve.

We thus proceeded to establish an *in vitro* system where formation of a multilayered epithelium is triggered by long term Ca^{2+} -treatment of confluent keratinocytes thus resulting in a multi-layered epithelium (Vaezi et al., Dev. Cell, 2002). Importantly, in this system TJs are formed in the most apically localized cells that no longer directly interact with the ECM through integrins, thereby mimicking the *in vivo* situation. Regrettably, as these cells are now also tightly interconnected by intercellular junctions at the lateral but also at the basal surface, quantification of vertex displacement was extremely complicated due to complex patterns of force dissipation in this system. We therefore used atomic force microscopy (AFM)-based force indentation spectroscopy to quantify apical cortical stiffness, which we as others before show to directly depend on tension generated by the actomyosin network (new Supplementary Fig. 5i) and asked whether E-cadherin would also control tension and global contractility in the absence of integrin-ECM contacts in a TJ-forming multilayered system. These data revealed that loss of E-cadherin reduced the actomyosin tension-dependent apical stiffness in these cells (new Fig. 5g-i), thus linking the mechanical and barrier properties of this layer. Most importantly, both reduced barrier function (Fig. 8e) and stiffness (new Fig. 8f, Supplementary Fig. 8b) were rescued by inhibition of EGFR, providing further molecular insight into how E-cadherin controls tension, organization of the cell cortex and restricted barrier formation.

2) What accounts for increased vinculin staining at uppermost layers? Vinculin is recruited at junctions upon mechanical tension but is it also generally stabilized? Or is it under transcriptional control?

Otherwise one would not expect an expression intensity "gradient", only a difference in localization (AJs vs. cytoplasm).

We did not detect differences in total protein expression of vinculin upon loss of E-cadherin, neither *in vivo* epidermis (Supplementary Fig. 3b) nor *in vitro* (Supplementary Fig. 5e,f), despite the observed strong reduction in junctional vinculin in the SG2 layer. Furthermore, no changes in total vinculin levels were observed during epidermal barrier morphogenesis, neither before (E15.5) intercellular recruitment of vinculin and SC barrier formation nor after junctional recruitment of vinculin (E16.5) that coincides with barrier formation (new Supplementary Fig. 4b,c). Together, these results show that E-cadherin does not control vinculin protein expression levels and indicate that recruitment of vinculin to AJs is most likely due to changes in its localization.

3) The discrepancy between the ZO-1 localization and cell-cell adhesion patterns in control vs. vinculin null keratinocytes in vitro vs. in vivo (ZO-1 pattern) is confusing. A PKC-dependent translocation mechanism of ZO-1 from TJs to focal adhesions has been described. Are the ZO-1-positive patterns in vinculin^{-/-} keratinocytes integrin-dependent? Do they co-localize with integrins instead of E-Cad in vinculin null keratinocytes?

We used ZO-1 staining to show how loss of either vinculin or E-cadherin differentially affect early AJ zipper formation and recruitment of F-actin, as this antibody has a much better signal to noise ratio than e.g. P-cadherin. However, we see the very same difference between the different mutant cells using P-cadherin as marker. It is well known that in many cultured simple epithelial cells ZO-1 is recruited to initial cadherin positive AJ zippers (Niessen & Gottardi, *Biochim. Biophys. Acta*, 2008) and only translocates to tight junctions later when junctions are more mature resulting in the formation of a functional barrier. This recruitment of ZO-1 to early AJ zippers is also true for ZO-1 in primary control keratinocytes during early intercellular junctions formation (now Fig. 5b). These ZO-1 positive zippers co-stain with E-cadherin and beta-catenin (not shown) and also resemble staining for vinculin in control cells. Nevertheless, as the reviewer has a valid point, we now co-stained for ZO-1 and paxillin, as a marker for focal adhesions, but could not detect significant co-localization in both control and Vinculin^{-/-} keratinocytes, indicating that ZO-1 does not accumulate in focal adhesions. These data are now included as new Supplementary Fig. 6f.

4) What about the P-cadherin/E-cadherin zipper-patterns in in vitro keratinocytes? Are those perhaps integrin-dependent? Are they differentially sensitive to changes in substrate stiffness (might be linked to the possibility that SG2 layer could be in contact with more rigid Stratum Corneum).

The reviewer touches upon a very interesting point and other laboratories have already shown that the ability of AJs to respond to force is interdependent on integrins see e.g. (Muhamed et al., *J. Cell Sci.*, 2016). Moreover, cells were shown to be more contractile on stiffer substrates (Engler et al., *Cell*, 2006; Ng et al., *J. Cell Biol.*, 2012), and increased contractility can also directly influence intercellular adhesive contacts (De Rooij et al., *J. Cell Biol.*, 2005). However, as focal adhesion-like cell-ECM contacts are not present in the SG2 layer we did not further pursue this question. See also answer to reviewer 1.

We thank the reviewer for pointing out that the rigidity of the apical stratum corneum provides a possible explanation for the higher junctional tension, as suggested by increased α 18 and vinculin staining in the SG2 cells. Alternatively, the differential distribution of adherens junctions in SG2 cells versus lower suprabasal cells may explain the difference in tension, as lower cells have E-cadherin positive adherens junctions at all intercellular interfaces whereas in SG2 cells adherens junctions are found on the basolateral interfaces, mediating contacts with cells below and at the lateral position but are no longer present between SG2 and SG1 cells, likely resulting in an altered distribution of tension. In

agreement, the *in vitro* multi-layered sheets do not form a SC but still restrict TJs to the most apical layer suggesting that the stiffness of the SC is not required to induce junctional polarization and favoring a model in which mechanical anisotropy induces differential stabilization of junctional components in the barrier-forming layer. We now discuss these two potential models in the discussion section.

5) The role of EGFR: The staining for EGFR in fig 7. is of suboptimal quality. It is not clear if EGFR in the uppermost SG layer is intracellular (endocytosed => active EGFR-signaling?) or apical (mislocalized)? Which cells produce the most EGF or do they all produce it?

We apologize for the suboptimal image and have now improved this by adding merged images of ZO-1 and EGFR, as well as single images showing enrichment of EGFR at sites of ZO-1 positive TJs (Fig. 7a).

Although we observe potential vesicular staining with EGFR antibodies, this staining, unlike the junctional signal, was still present when we used sections of epidermal EGFR^{-/-} mice (kind gift of Francesca Macia and Stuart Yuspa, NIH) to validate the specificity of our EGFR staining, making it hard to carefully assess this difference as some of the signal is likely to be background. To better address this question we performed cell surface biotinylation experiments in primary keratinocytes treated with Ca²⁺ to induce TJ formation (48h) and found increased internalization of the EGFR upon loss of E-cadherin. This was restored to control levels upon inhibition of the EGFR, indicating that loss of E-cadherin activates EGFR signaling, leading to increased endocytosis as suggested by the reviewer (new Fig. 8c,d).

At present it is not clear which cells in the epidermis produce the most EGF. Several other EGFR ligands are also expressed in the epidermis (Schneider & Yarden, *Semin. Cell Dev. Biol.*, 2014), further complicating the pinpointing of the definitive ligand. As some evidence suggest that EGFR can be activated independently from ligand (see e.g. (Lambert et al., *J Invest Dermatol*, 2006; Wang et al., *Proc. Natl. Acad. Sci. U. S. A.*, 2012)), and more specifically interactions between E-cadherin complexes and the EGFR may directly control EGFR activity (Curto et al., *J. Cell Biol.*, 2007), we can at present not even rule out whether E-cadherin dependent regulation of EGFR activity depends on EGFR ligand binding or not. We tried very hard to identify a possible interaction between EGFR and E-cadherin complex, but were thus far unsuccessful (see also discussion). However, we found that tension directly controls EGFR activity (new Fig. 8g), perhaps suggesting a ligand independent mechanism. Further molecular investigations are an interesting future goal.

6) What happens to junctional components in in vitro cultured normal and ECad-null keratinocytes upon addition of EGF? Are they endocytosed together with EGFR?

Using cell surface biotinylation experiments, we observe an increase in internalization of the TJ protein occludin, but not claudin-1 or P-cadherin (not shown), in E-cadherin^{-/-} compared to control, which was reversed by the EGFR inhibitor genitib (new Fig. 8d). As already mentioned, this was accompanied by an increase in EGFR internalization. However, we were unable to detect any colocalization of occludin and EGFR in vesicles.

Minor comments:

- *An open question is where the force comes in the uppermost layer? ECad is found in all layers but why only uppermost layer forms contractile TJs? Could mechanical properties of the overlaying Stratum Corneum play a role (SC has been suggested to be stiffer than the viable layer perhaps due to loss of water that might cause the SC layer to "shrink")? Or is it anticipated that the tension is due to "stretching" of less proliferating squamous top cell layers by the pressure from basal layers proliferating and pushing upwards? These issues should be shortly discussed in order to better describe how the*

findings in this study come together to support the proposed model.

We thank the reviewer for pointing this out and now discuss in more detail the potential models that may explain how junctional tension is differentially regulated. One possibility is that the stiffness of the forming stratum corneum is the driving force for increased AJ tension and TJ formation only in the SG2. Alternatively, the anisotropy in junctional distribution in the SG2 (AJs are only basolateral but not apical between SG2 and SG1 cells, whereas spinous layer have actin connected AJs junctions at all intercellular interfaces), may explain the differential AJ composition and exclusive TJ formation in the SG2. Please also see discussion above.

• *The underlying logic is difficult to follow in discussion of the model that incorporates the observations that both E-Cad mediated stimulation or strong inhibition of EGFR compromise TJ formation into the findings in E-Cad null epidermis. Especially since more modest inhibition of EGFR in ECad^{-/-} keratinocytes instead facilitated TJ-formation? How do these come together? Moreover, in simple epithelium EGF appears to increase TER (for example the reference "Epidermal growth factor receptor activation differentially regulates claudin expression and enhances transepithelial resistance in Madin-Darby canine kidney cells." Singh AB et al. J Biol Chem. (2004) and others). Why should this be the opposite in keratinocytes? This could maybe be rephrased in the discussion to clarify the links between the observations.*

We apologize for the somewhat confusing discussion and have tried to more clearly discuss the data and extract a model from this data.

Our data indicate that E-cadherin-dependent tension regulation is required to spatiotemporally constrain EGFR activity in the SG2 to control organization and thus tension of the actomyosin cortex to stabilize and facilitate maturation of TJs. Together, the *in vitro* and *in vivo* data suggest a two-step model in which E-cadherin actively inhibits premature TJ formation in lower epidermal suprabasal layers while promoting TJ formation in the SG2 layer (represented in new Fig. 8h). In lower layers, all intercellular interfaces contain E-cadherin AJs that are under low tension and stimulate moderate EGFR activity, thus further reducing cortical tension and destabilizing TJs by promoting internalization. In contrast in the SG2 layer, AJs are absent from the apical intercellular interface, resulting in an anisotropy of actomyosin force distribution that may be sufficient to increase tension on junctions allowing for increased unfolding of α -catenin and vinculin. This in turn reduces EGFR activity thus increasing cortical stiffness while at the same time stabilizing TJs at the cell surface. Unfortunately, we were unable to properly detect active EGFR in the skin, either using phospho-EGFR antibodies or MAP-kinase antibodies, despite using a range of conditions. As our data indicate that E-cadherin controls both the localization and activation status of the EGFR and that the timing of activation may be important, it will for the future be essential to develop tools that allow precise monitoring of EGFR activity and localization.

Some literature, such as the above-mentioned papers, indicate that EGF can stimulate TJ formation. whereas several reports describe that EGF inhibits TJ function (see e.g. Von Itallie et al., 1995; Ikari et al., J. Cell. Physiol., 2011). For example, in e.g. ovarian cancer cell lines addition of EGF results in a downregulation of claudins and a decrease in TJ function as measured by TER (Ogawa et al., Histochem. Cell Biol., 2012). The different observations that EGF or some of its downstream mediators can either promote or inhibit TJ function perhaps further illustrates that spatiotemporal timing may be essential. These studies were all done in simple epithelial cells that form a monolayer instead of a multilayered epithelial sheet, and none addressed timing of activation. We therefore did not discuss these studies in detail.

- *Not clear whether and how much EGFR is enriched at SG2 - compare fig7 and Fig 8a!*

Figures 7 and 8A in the previous version of the manuscript described different time points. Fig. 7 shows newborn skin in which the EGFR is enriched at TJs, as quantified in Fig. 7b. In old Fig. 8a (now new Fig. 7g,i) EGFR is shown at E15.5 and E16.5 and reveals that upon initiation of epidermal barrier formation (E16.5), EGFR translocates to suprabasal junctions, including tight junctions but at this developmental time point is not yet enriched in these junctions.

- *On page 9 line 281 the reference to (Fig. 6f,g) now says 5f,g*

- *References have some issues with symbols (at least Bueler 2013, Shen 2005 and Yonemura 2011)*

We thank the reviewer for pointing these mistakes in figure and literature references out and made corrections.

Reviewer #3

This paper addresses the question whether E-cadherin-mediated force transduction in the epidermis is important for epithelial differentiation and tight junction formation. Using a knockout approach, E-cadherin deletion is shown to lead to reorganization of the actin cytoskeleton and components known to be involved in force transduction such as vinculin. As E-cadherin is known to lead to a defect in tight junction formation in the skin and they find that force transmission is affected in cultured keratinocytes, the authors postulate that mechanical force controls the formation and positioning of tight junctions in the epidermis. They further conclude that the tissue distribution of EGFR is also deregulated and that E-cadherin may regulate junction formation by controlling the timing of EGFR signaling.

Adherens junctions are major transducers of mechanical force between neighboring cells. E-cadherin is in many epithelia the main cell-cell adhesion protein of adherens junctions and is a component of the force-transducing molecular linker. Interplay between EGFR signaling, E-cadherin and intercellular junctions has been analyzed in different systems and is known to be a key mechanism of epithelial differentiation and junction formation. However, little is known about these mechanisms in vivo and, in particular, the role they play in the differentiation and function of stratified epithelia. The paper thus addresses an important and timely question.

The paper presents a considerable amount of high quality data that point to potentially exciting conclusions. However, much of the presented evidence is indirect and incomplete. Moreover, sometimes it is difficult to reconcile the actual figures with the description in the text. Hence, the main conclusions are not well supported.

We thank the reviewer for appreciating our study and the quality of the data and have now done substantial additional experiments to clarify and further mechanistically connect the data. In brief, these new data show that E-cadherin dependent regulation of tension constrains EGFR receptor activity to restrict TJ formation to the SG2 layer. We apologize that the reviewer thought our descriptions were unclear and worked extensively to improve this. Please see below for our answers to the specific comments.

1) A major drawback of the experimental approach is the lack of experiments directly measuring tension

changes in the epidermis. While the staining of specific alpha-catenin epitopes and vinculin may provide indications of tension alterations, they do not provide a direct readout for tension. Admittedly, some methods used to measure tension may not work in vivo and in stratified epithelia (e.g., ablation experiments) and others are cumbersome to implement (tension sensors). However, to substitute such measurements with an in vitro method that makes use of isolated pairs of cells to measure tension is inappropriate. In fact, some of the results shown here quite clearly indicate that the in vitro and in vivo systems do not yield the same results. Instead of traction force microscopy, which is difficult to implement with monolayers, one would expect at least in vitro force measurement using continuous layers of cells (e.g., laser ablation or tension sensors).

We agree with the reviewer that unfolded α -catenin as well as vinculin are not a direct measurement of tension but only serve as indicators of increased tension at adherens junctions. We used traction force microscopy on adhesive doublets of cells to show that E-cadherin loss directly affects intercellular tension. We consider this an essential point, especially as these doublets form desmosomes, which are considered the main and crucial mechanical intercellular adhesive systems in keratinocytes. Nevertheless, we fully agree that these conditions in which we measured tension do not fully recapitulate conditions that represent the multilayering and junctional distribution of the epidermal tissue.

Currently available tension sensors e.g. vinculin, E-cadherin (Grashoff et al., Nature, 2010; Borghi et al., Proc. Natl. Acad. Sci., 2012) measure tension across specific single molecules, which does not necessarily correlate or read out the tension across the adhesion structure. Direct indications for tension at junctions *in vivo* come from junctional laser ablation studies on simple epithelia in Drosophila and Zebrafish (Fernandez-Gonzalez et al., Dev. Cell, 2009), using recoil at tricellular junctions as a junctional tension read out. However, these epithelia are not encased by a protein-lipid cross-linked layer, such as the stratum corneum of the epidermis. We still put considerable effort into trying to set up laser ablation experiments *in vivo* on the epidermis of both E16.5 and newborn embryos. However, the physical properties of the stratum corneum were indeed a major barrier for efficient laser ablation of intercellular contacts in the SG2 layer. We tried several different lasers (UV or far-red two photon) and microscope systems but were unable to achieve *in vivo* ablation of intercellular junctions.

We then turned to *in vitro* laser ablation and used primary keratinocytes that were allowed to form a multilayered epithelial sheet (Vaezi et al., Dev. Cell, 2002) where the ZO-1-containing junctions are formed at the apical side of the most apical layer, which is not in direct contact with the substrate, thus mimicking the *in vivo* SG2 layer situation. Regrettably, as these cells are tightly interconnected by intercellular junctions not only at the lateral but also at the basal surface, quantification of vertex displacement was extremely complicated due to complex patterns of force dissipation in this system. Please also see also answers to reviewers 1 and 2.

To show that E-cadherin directly affects mechanics of the actomyosin cortex, we then went on to quantify apical cortical stiffness of the TJ-forming cell layer by setting up atomic force microscopy (AFM)-based force indentation spectroscopy. These experiments revealed that E-cadherin^{-/-} keratinocytes show decreased cortical stiffness in the apical layer (new Fig. 5g-i). We further confirmed using blebbistatin that this cortical stiffness depends on myosin-generated tension (new Supplementary Fig. 5i). Thus, our *in vivo* observations that show an increased junctional vinculin and α -18 in the TJ forming SG2 layer together with the TFM and new AFM data in a multilayered epithelium collectively indicate that E-cadherin controls the mechanical properties of the TJ-forming layer by regulating tension across junctions and the mechanical coupling of the actomyosin cortex.

2) As the vinculin deletion results do not support the model of tension and tight junction positioning, it is surprising that these experiments were not completed by measuring the effect of vinculin depletion on

cell-cell tension using the cultured keratinocytes. Based on the data as they are, one could also speculate that the vinculin deletion results mean that tissue tension is irrelevant for tight junction organization and epidermal differentiation.

We thank the reviewer for pointing this out. Using the above-mentioned AFM force spectroscopy approach we measured cortical stiffness of the TJ-forming apical layer in Vinculin^{-/-} multilayered keratinocytes. In line with our *in vivo* observation that vinculin mice are viable and do not show a defect in basal to apical junctional tissue polarization in the epidermis, we found that Vinculin^{-/-} multilayered sheets showed no decrease in cortical stiffness (new Fig. 6h, Supplementary Fig. 6g,h). Instead, the data reveal a small, albeit statistically non-significant trend for increased stiffness compared to control cells. This result confirms that vinculin is not essential for AJ mediated force transduction in keratinocytes, and suggests that vinculin might function to balance forces across AJs.

3) It is unfortunate that only ZO-1 is used as a marker for tight junctions, as it is clearly not specific for that junction in the epidermis. It is often not clear why some areas positive for ZO-1 are marked as being tight junctions and others are not. For example, in Figure 4e, the image at E16.5 shows staining of all cell layers by ZO-1. Some areas are declared as tight junctions but without any further support for such a conclusion. The image clearly illustrates that it is not sufficient to stain for ZO-1 to identify tight junctions. More specific markers should be added to the analysis, preferably membrane proteins such as occludin.

We agree with the reviewer that ZO-1 is not completely specific for TJs even though it is enriched in TJ structures in the SG2 layer of newborn epidermis (see e.g. Fig. 1e-f) and recently was shown to label functional, occluding TJs in mouse epidermis (Yokouchi et al., eLIFE, 2016). Claudin-1, although also enriched at TJs, is also found at sites of cell-cell contact in lower layers. We therefore used occludin as a more specific marker for SG2 (Tunggal et al., 2005) to stain newborn whole mount epidermis. Similar to ZO-1, occludin distribution is no longer continuous in the SG2 layer upon loss of E-cadherin and becomes unevenly distributed at sites of cell-cell contacts in lower spinous layers (new Fig. 3d; new Supplementary Fig. 3e?) These clusters co-stain with ZO-1, thus indicating that loss of E-cadherin disturbs TJ structures in the SG2 and promotes the formation of premature dysfunctional TJ like structures in the spinous layer.

4) Fig. 1b - In contrast to the description in the text (page 4), there seems to be a considerable amount of vinculin staining in layers lower than SG2. Where one can see E-cadherin, there is vinculin. Similarly in Fig. 3, phalloidin in spinous layer: It seems an overstatement to describe the right panel as a 'strongly enhanced cortical' staining. It is hardly visible. Moreover, the vertical section through the layers underneath seems to suggest that the phalloidin staining is generally stronger also in the SG layers. Here too, staining a more specific tight junction marker would considerably strengthen the conclusion that E-cadherin deletion leads to redistribution of tight junction proteins. Panel e seems to suggest a surprisingly small effect on ZO-1 and whether that alone would be sufficient to lead to a loss of function is questionable.

We apologize that the images shown were not completely clear, whereas careful quantification of the data in Fig. 1d,f indicate a robust difference. Quantification of vinculin revealed a four-fold enrichment in the SG2 layer compared to the lower layers. We now present the panels for vinculin and actin as a grayscale to more clearly illustrate the differences.

For the quantification of cortical F-actin we used the very intense phalloidin staining of scattered SG1 cells, as a maximum intensity, making differences between SG2 and spinous layers look relatively small compared to the difference with SG1. The important point for loss of E-cadherin is not to compare the cortical F-actin in spinous versus granular layers in E-cadherin^{epi/-} mice but to compare the difference in

intensity between the spinous layers of control and E-cadherin^{epi-/-} mice. The images and quantification clearly show that loss of E-cadherin induces an increase in cortical F-actin in this layer (Fig. 3e). We have toned down the conclusions for F-actin to state “enhanced” instead of “strongly enhanced” to avoid overstatement in case of actin.

Finally, as already mentioned above, we also added whole mount stainings for the TJ protein occludin to further provide evidence for the E-cadherin induced alterations in TJ protein localization both in the granular layer and spinous layer. Importantly, although overall intensity in SG2 for ZO-1 (Fig. 3f) or occludin (not shown) is not altered when comparing the SG2 layer from control and E-cadherin^{epi-/-} mice, the staining for both occludin and ZO-1 is no longer continuous but shows many gaps (new Fig. 3d), indicating that TJs no longer form a continuous seal and thus explaining the in vivo loss of TJ epidermal barrier function upon loss of E-cadherin, which we reported in Tunggal et al, EMBO J. 2005.

5) Page 5, lines 138/9 - While there is clearly vinculin staining in the proximity of tight junctions, these images do not offer the resolution to demonstrate that this is tight junction associated vinculin. Figure 1 shows a similar distribution of E-cadherin and a complete overlap with vinculin in SG2. How is that be compatible with vinculin in tight junctions?

We completely agree with the reviewer that vinculin co-localizes with E-cadherin/AJs and not with tight junctions, and apologize that this conclusion was not coming across clearly. We now more clearly state that Vinculin co-localizes with E-cadherin and that these vinculin/E-cadherin positive AJs only partially overlap with TJ.

6) Page 8, figures 4 and supplementary 5 - The increase in Rac activity is not convincing as there seems to be an increase in total Rac as well. One wonders why only Rac is investigated as RhoA may directly lead to tension.

We have in fact also performed Rho activity assays and could not detect significant changes in Rho activity. We initially chose to include a representative experiment on the Rac data, as quantification of these data, taking into account total Rac levels, showed a decrease in Rac and this small GTPase was shown to be activated upon cadherin engagement and organizes junctional F-actin (Noren et al., J. Biol. Chem., 2001). However, our new experiments further indicate that exact timing of the GTPase activities is likely to be crucial. Therefore, we feel that direct inhibition of myosin II provides more clear evidence of actomyosin tension in TJ assembly and decided to remove the Rac data from the manuscript and not include the Rho data.

7) Why did the authors choose to investigate only ppMLC2 phosphorylation and not also single phosphorylation (the text confuses these two different forms). Based on supplementary figure 5f, there does not seem to be any active myosin at the junction/membrane even in controls, which seems surprising. How do the authors envision that cortical tension is generated? How is the distribution of ppMLC and pMLC in vivo?

We thank the reviewer for pointing this out and have clarified the text accordingly. We have investigated the status of phosphorylation on myosin light chain Thr18/Ser19 as phosphorylation of both sites is considered to induce contraction of actomyosin fibers. In contrast, single phosphorylation of Ser19 is proposed to mediate organization and stability of myosin but not necessarily to indicate its contractile status (Ikebe et al., J. Biol. Chem., 1988; Watanabe et al., Mol. Biol. Cell, 2006). We therefore chose to assess the ppMyosin in keratinocytes.

Visualization of the cortex in these cells using light microscopy is not trivial as it is very thin (100-200 nm), explaining why the majority of ppMyo immunofluorescence signal comes from actin bundles and stress fibers. However, the cortical actin fibers that are aligned in parallel to the intercellular junctions, are highly decorated with ppMLC2 and are radially connected to AJ zippers. A previous report showed that formation of these contractile cortical actin fibers and AJ zippers are interdependent and specifically depend on myosin activity (Vaezi et al., Dev. Cell, 2002). Furthermore, our AFM data demonstrate changes in cortical stiffness upon loss of E-cadherin, whereas cortical stiffness itself depends on myosin-generated tension (new Fig. 5g-I; Supplementary Fig. 5i).

Unfortunately, in our hands staining for ppMyosin did not properly work in immunohistochemistry or whole mounts despite extensive testing of different antibodies and a range of conditions. The only antibody that gave a specific signal on tissue sections was directed against phosphor-serine 19 Myosin (pMyosin) Importantly, this staining showed enhanced pMyosin cortical localization in the granular layer (new Supplementary Fig. 2c).

8)page 8: the authors seem to ask whether abolition of tension leads to tight junction reformation by inhibiting MLCK. This experiment would be far more conclusive if myosin were inhibited directly with blebbistatin. Why was only MLCK studied and not the RhoA/ROCK pathway?

We completely agree with the reviewer. Our initial hypothesis was that aberrant contractility induced by loss of E-cadherin accounted for the inability to form a proper TJ barrier. To test this we assessed if inhibition of myosin signaling using ML7, blebbistatin or Y-27632 would rescue TJ function in E-cadherin^{-/-} keratinocytes. However, we did not observe a rescue of TJ function in the E-cadherin^{-/-} keratinocytes with any of the inhibitors. Of these, we chose to include the ML7 experiment shown in the supplementary data as it was the most comprehensive data set. Our new data lead us to revise this hypothesis and to postulate that E-cadherin-mediated tension is required for TJ formation. To test this directly we inhibited myosin II by blebbistatin in control cells, as also suggested by the reviewer, and observed that blocking myosin indeed prevents TJ function. We have therefore removed the ML-7 data from the paper.

Minor comments:

1) Fig. 1c - The alpha18 antibody staining should be shown as an individual channel so that the gradient in staining becomes more evident.

We thank the reviewer for this suggestion. Next to the quantification that was already included (Fig. 1d) we now also have added the single channels of the XZ in the Supplementary figure 1b.

2) Fig. 1g - The term 'subapical' is confusing. It seems that the authors are referring to lateral junctions. This term normally refers to an epithelial region between tight junctions and the apical domain.

We thank the reviewer for pointing this out and apologize for inappropriately using this term. We have replaced this term with "lateral junctions".

3) Page 5 - The conclusions of the first results section (lines 115-126) should be phrased more carefully. The experiments reported used indirect markers for cell-cell tension and no actual tension experiments are shown. The data are supportive of the described tension gradients but do not prove them.

We have revised the text to more precisely describe the data and acknowledge that the markers used are indirect markers for tension.

4) Page 6, line 180 - This statement reads as though loss of E-cadherin would lead to formation of functional tight junctions elsewhere in the skin; however, I don't assume this is what the authors intended to conclude. The data just suggest a minor redistribution of ZO-1 compatible with the previously **reported** loss of functional tight junctions in such animals.

We thank the reviewer for pointing this out. We have rephrased this sentence to make the discussion more clear, as we indeed do not want to give the impression that functional TJs are seen in lower layers upon loss of E-cadherin. What we see are scattered clusters that are positive for occludin and ZO-1, suggesting premature recruitment of TJ components to non-functional structures, as previously shown by biotin penetration assays (Tunggal et al, EMBO J. 2005).

5) Figure 4/page 7 - The section makes a statement about functional barrier formation. Where is it shown whether the barrier is functional or not? Do the authors rely on previous publications?

We rely on previous publications from others and us as well as own extensive unpublished experiments that show, using dye exclusion assays, that SC barrier formation is initiated at E16.5 at the dorsal side of the embryo see e.g. (Hardman et al., Development, 1998; Tunggal et al., EMBO J., 2005; Koch et al., J. Cell Biol., 2000).

The point we wanted to make is that the polarization of junctions and the first signs of a TJ in the stratum granulosum coincide with the first appearance of a functional stratum corneum barrier, which has not been documented yet in the literature. We have now rephrased this section to make this more clear.

6) Fig. 8a/b - this is really a central figure supporting the main conclusion and should hence be quantified.

We have now quantified the recruitment of EGFR to suprabasal layer and this shows an increased and premature recruitment of the EGFR in E15.5 embryos (new Fig. 7i).

7) Fig. 8c - Why are the Ecad-/- cells incubated with DMSO and not like the controls?

We thank the reviewer and apologize for this mistake in labeling. None of the cells in this experiment (now new Fig.8a) were treated with DMSO since the EGF that was used here is dissolved in normal medium.

8) The discussion concludes that a mechanical circuit was identified that integrates adhesion, cortical contractility and chemical signaling etc. The paper does not contain any evidence for tension regulating EGFR signaling or for altered cortical contractility (which would require altered myosin activity at the cortex).

We have now added extensive new data that show that tension regulates EGFR signaling (new Fig. 8g), (which has also been observed by others in simple epithelia) and tight junction formation (new Fig. 5k). Most importantly, we find that E-cadherin through regulation of EGFR activity not only controls tight junction stability (new Fig.8c,d) and function (Fig. 8 a,b, e) but also the mechanical properties of the actomyosin cortex (new Fig. 8f). We have integrated these new data and more carefully discuss the findings as well as phrasing of our conclusions. Collectively these data put forward a two-step model in which E-cadherin integrates mechanical and chemical signaling to control the spatial formation of a TJ barrier (new Fig.8h). This model is now extensively discussed in the discussion section (also see answers to reviewers 1 and 2).

Reviewers' comments:

Reviewer #1 (Remarks to the Author):

In this resubmission, the authors now include substantial new data that address several of the significant issues raised in the critiques. The major strength of this study is its basis in whole mount tissue sections obtained from genetically altered mice, underpinned by the extensive knowledge of epidermal cell biology. But the focus on the SG in the intact epidermis is also the major challenge that makes teasing out complex mechanisms particularly difficult.

Their efforts to address the critiques are substantial, and the resulting studies were carefully done, to the limits of their experimental and biochemical abilities. The traction measurements are now clear and convincing, and they clarified questions regarding P- and E-cadherin. Immunostaining images are also improved. Although efforts to assess tension in the whole mounts were unsuccessful, some of the *in vitro* studies helped clarify links between perturbations and altered tension.

There are clear correlations between E-cadherin expression, EGFR activity, cortical tension, vinculin accumulation, tight junction organization, and barrier function. However, there are also several unresolved issues that preclude pulling together a clear mechanism.

A significant overarching issue is that, despite their substantial efforts, the current data and unresolved questions do not yet support their rather strong, mechanistic assertions in the title, abstract and discussion.

Main points:

The evidence that EGFR, E-cadherin expression, and tension appear intimately coupled. The study still begs the question as to what is regulating tension, which includes contributions from both cortical actin organization and global cell contractility (and possibly keratin). Inhibiting myosin II reduces tension and increases EGFR activity (in wild type cells), and EGFR inhibition in E-cadherin $-/-$ cells increases cortical tension measured by AFM (response to comment 5, Reviewer 1) and TJ barrier restoration, independent of E-cadherin. These data expose an unresolved feedback mechanism that correlates EGFR activity with tension. But pertinent to the title, it is not clear that this link actually involves E-cadherin-mediated mechanotransduction, implied in the title.

The increased Rac and undetected RhoA signaling is puzzling, particularly in light of myosin II phosphorylation in regions of actin fibers at AJs. Rac typically reduces tension. What is activating myosin II, and how is this related to changes in EGFR activity? Do they observe increased Rho activity in tissue culture? It could very well be the case that signaling on flat tissue culture plastic may differ from the *in vivo* context, but at this point this connection between EGFR and cortical tension is still missing. But this issue is important because this pathway affects TJ's and tension.

The authors suggest that the reason for the apparent discrepancy between Muhamed et al and their findings was the use of single cells. However, Collins et al (Curr Biol 2014) identified a very similar mechanotransduction pathway in endothelial cells that requires VE-cadherin and vascular endothelial growth factor receptors and activates an integrin-RhoA pathway (also different from this present manuscript). Coon et al (JCB, 2015) showed that VE-cadherin association with VEGFR2 and VEGFR3 is required to form the core mechanosensory complex *in vitro* and *in vivo*. A much more likely explanation for the different observations in epidermis is the nature of the mechanical input and the possible influence of other adhesion proteins such as P-cadherin and integrins that may have different effects in different tissue contexts (see Wang et al PNAS, 1998 p14821).

Related to the above question and to one of the other Reviewers' points, integrins were not

considered in this study. What role are integrins playing? The authors addressed this by focusing on focal adhesions and paxillin staining, but integrins can be activated by intracellular signals as well as by secreted extracellular matrix via mechanisms that do not involve focal adhesion maturation. Rosario et al (2009) Dev Biol showed that secreted fibronectin affected cell polarity and cell movements in *Xenopus* embryos, and other reports implicated secreted laminin in integrin-dependent autocrine signaling.

The formation of mature desmosomes in their cultures and in vivo does raise the question of the role of keratin in the cortical tension measurements. If desmosomes require E-cadherin, then E-cadherin would indirectly regulate keratin filaments, which also likely contribute to cortical stiffness. They mention keratin in the response, but did not specifically address this.

Reviewer #2 (Remarks to the Author):

The revised manuscript by Rübsem et al. has been significantly strengthened by the addition of data on the mechanical properties of in vitro cultured multilayered primary keratinocytes. With this model they have been able address many of the remaining gaps in their model. Together with other added data they now provide more convincing mechanistic links connecting Ecad, mechanical tension and EGFR-axis. The revised manuscript now more clearly describes their suggested model. Also all the other main concerns of this reviewer have been addressed sufficiently. While several interesting new questions are raised by these findings I feel that they are subject to further studies and beyond the scope of the current manuscript. The revised manuscript now provides evidence for a novel functional model describing E-cadherin/tension/EGFR-dependent molecular machinery, that is required to position tight junctions to the most suprabasal viable layer in the epidermis. These results will likely be of interest for scientists on several related research fields and are therefore suitable for publication in Nature Communications.

Reviewer #3 (Remarks to the Author):

The authors addressed all of my comments and added convincing additional data.

We would like to thank all three reviewers for their appreciation of our revision and were happy to see that both reviewers 2 and 3 had no further comments and were fully satisfied with our new data and revised manuscript. We would further like to thank reviewer 1 for his/her additional critical input.

The reviewer felt that some of our conclusions were too strong. We therefore went carefully through the manuscript and took extra care to rephrase central conclusions of the manuscript to ensure that they were precise and accurately reflecting the data. Some points were also previously raised by reviewer 2 and 3, as is in part also reflected by the answers below.

Reviewer 1

In this resubmission, the authors now include substantial new data that address several of the significant issues raised in the critiques. The major strength of this study is its basis in whole mount tissue sections obtained from genetically altered mice, underpinned by the extensive knowledge of epidermal cell biology. But the focus on the SG in the intact epidermis is also the major challenge that makes teasing out complex mechanisms particularly difficult.

Their efforts to address the critiques are substantial, and the resulting studies were carefully done, to the limits of their experimental and biochemical abilities. The traction measurements are now clear and convincing, and they clarified questions regarding P- and E-cadherin. Immunostaining images are also improved. Although efforts to assess tension in the whole mounts were unsuccessful, some of the in vitro studies helped clarify links between perturbations and altered tension.

There are clear correlations between E-cadherin expression, EGFR activity, cortical tension, vinculin accumulation, tight junction organization, and barrier function. However, there are also several unresolved issues that preclude pulling together a clear mechanism.

A significant overarching issue is that, despite their substantial efforts, the current data and unresolved questions do not yet support their rather strong, mechanistic assertions in the title, abstract and discussion.

We thank the reviewer for appreciating our substantial efforts to address the reviewers critique and the strengths of the study. As the reviewer acknowledges, the fact that TJs are only found in the SG2 in intact epidermis is both an important and interesting biological question but also a major experimental challenge. Nevertheless, by combining in vivo and in vitro models we have made major

inroads in furthering our understanding of how multi-layered epithelia, such as the epidermis, restrict the formation of barrier-promoting TJs to the upper viable layer, a process that was not understood until now. Summarizing, our combined in vivo and in vitro data show that:

1. E-cadherin controls TJ function in vivo in the SG2 layer (Tunggal et al., 2005) and in cultured multi-layered epithelia (Figure 5j).
2. E-cadherin restricts formation of occludin/ZO-1 positive TJ like structures (Figure 1,3,4) to the SG2: loss of E-cadherin results in punctate like structures positive for ZO-1, Occludin (Figure 3) and claudin-1 (not shown) at intercellular contacts in lower layers.
3. E-cadherin restricts formation of vinculin-positive, α -18 positive (indicative of unfolded α -catenin, previously shown by others to be regulated by tension) to the SG2 layer (Figure 1 and 3A). Moreover, these vinculin high, α -18 positive junctions correlate with increased organization of cortical F-actin in the SG2 (Figure 2). Based on many in vitro studies by others that showed that unfolding of α -catenin and subsequent recruitment of vinculin strengthens junctions (e.g. Imamura et al. 1999, Le Duc et al. 2010, Yonemura et al. 2010), these results, although indirectly, indicate that AJs in the SG2 are under higher tension than those found in lower layers.
4. In agreement, E-cadherin controls intercellular junctional tension and tension distribution as measured by traction force microscopy (Figure 5d,e), and apical cortical tension in the upper layer of a cultured multi-layered epithelium (as measured by atomic force microscopy) (Figure 5g-i), which is the same layer in which TJs are formed in vitro.
5. E-cadherin regulates the localization and activity of EGFR in vivo (Figure 7) and in vitro (Suppl. Fig. 7)
6. Actomyosin contractility controls apical cortical tension, tight junction formation and EGFR activity in multilayered epithelia (Supplementary Fig. 5i, Figure 5k and 7g)
7. EGFR controls apical cortical tension and TJ formation downstream of E-cadherin in multi-layered epithelia (Figure 8).

Together, these data provide multiple causal links between E-cadherin, cortical tension, EGFR receptor and TJs and collectively provide substantial support to propose a model in which E-cadherin actively inhibits the premature formation of TJs as well as increased cortical F-actin organization in lower layers while promoting TJ stability by increasing cortical tension in the TJ layer and at the same time inhibiting EGFR.

Although we are confident about these findings, we fully agree with the reviewer

that some of the statements in the discussion and abstract were too strong and have therefore edited and toned down statements to ensure that they precisely reflect the experimental evidence and to avoid over-interpretation.

Nevertheless, we respectfully disagree on the reviewer's assessment regarding the title. We do not claim that E-cadherin regulates EGFR through tension by stating that E-cadherin mechanotransduction controls EGFR signaling. Instead we state that E-cadherin **integrates** mechanotransduction and EGFR signaling for which we provide extensive experimental evidence.

Main points:

The evidence that EGFR, E-cadherin expression, and tension appear intimately coupled. The study still begs the question as to what is regulating tension, which includes contributions from both cortical actin organization and global cell contractility (and possibly keratin). Inhibiting myosin II reduces tension and increases EGFR activity (in wild type cells), and EGFR inhibition in E-cadherin ^{-/-} cells increases cortical tension measured by AFM (response to comment 5, Reviewer 1) and TJ barrier restoration, independent of E-cadherin. These data expose an unresolved feedback mechanism that correlates EGFR activity with tension. But pertinent to the title, it is not clear that this link actually involves E-cadherin-mediated mechanotransduction, implied in the title.

As already summarized above, our initial data showed that E-cadherin controls both junctional and apical cortical tension as well as TJ assembly and EGFR. We further show that actomyosin contractility is required for apical cortical tension, as expected, but is also regulating EGFR activity and TJs. Finally, EGFR regulates TJs and apical cortical tension, thus revealing intimate links between E-cadherin, EGFR, cortical tension and the restricted formation of TJs. As the reviewer rightfully points out, our data indicates that E-cadherin controls actomyosin tension, EGFR and thereby TJs, and that EGFR activity also contributes to the regulation of apical stiffness.

To experimentally further address whether regulation of EGFR activity directly involves E-cadherin-mediated mechanotransduction, we have inhibited myosin activity with blebbistatin and show that myosin activity is required to regulate EGFR activity. To address how E-cadherin regulates this, we have rescued E-cadherin^{-/-} keratinocytes, which have higher EGFR signaling than control, with either full length E-cadherin or a C-terminal deletion mutant that can no longer interact with β - and α -catenin but still with p120^{ctn}. This mutant can thus not stably couple to

the actin cytoskeleton. Interestingly, both wt and E-cadherin mutant can reduce EGFR activity, suggesting that a direct, stable link of E-cadherin to the actin cytoskeleton is not necessary. In agreement with this finding, E-cadherin can directly interact with EGFR through its extracellular domain resulting in inhibition of EGFR (Qian et al., 2004). Moreover, a similar mutant in which the beta-catenin binding domain was deleted can still cluster and strengthen adhesion (Yap et al., 1998). As this mutant can still bind 120ctn, it also has the ability to regulate actin through interaction with e.g. N-Wasp (Rajput et al, 2013) or cortactin (Boguslavsky et al., 2007) and actomyosin controlled tension through regulation of Rho and Rac activity (see e.g. Ouyang et al, 2013; Epifano et al., 2013, Wildenberg et al., 2006), in agreement with our finding that inhibition of actomyosin results in increased EGFR activity. Taken together, a model to be tested in the future is that tension high adherens junctions stabilize the direct interaction of E-cadherin with EGFR allowing for effective EGFR inhibition. Alternatively, cadherins regulate actomyosin dependent tension directly to control EGFR activity, however, independently from engagement of cadherin/catenin complexes with the actin cytoskeleton.

The increased Rac and undetected RhoA signaling is puzzling, particularly in light of myosin II phosphorylation in regions of actin fibers at AJs. Rac typically reduces tension. What is activating myosin II, and how is this related to changes in EGFR activity? Do they observe increased Rho activity in tissue culture? It could very well be the case that signaling on flat tissue culture plastic may differ from the in vivo context, but at this point this connection between EGFR and cortical tension is still missing. But this issue is important because this pathway affects TJ's and tension.

The reviewer may have missed that the initial Rac data, showed a **reduction in Rac** upon loss of E-cadherin and not an increase. However, these data were no longer included in the first revision due to more conclusive experiments and observations made during the revisions that made these data redundant. We addressed this in the first revision in answering reviewer 2. We have in fact also performed Rho activity assays and could not detect significant changes in Rho activity. Importantly however, our new experiments included in the revision further indicated that exact timing of the GTPase activities is likely to be crucial. Therefore, we feel that direct inhibition of myosin II provides more clear evidence for the role of actomyosin in regulating tension, EGFR activity and TJ assembly. We fully agree with the reviewer that the Rac and Rho pathways are of great interest in light of our findings on tension, EGFR and TJs. However, this will require in depth, non-trivial carefully time-resolved experiments of these pathways, which is at present beyond the scope of this paper.

The authors suggest that the reason for the apparent discrepancy between Muhamed et al and their findings was the use of single cells. However, Collins et al (Curr Biol 2014) identified a very similar mechanotransduction pathway in endothelial cells that requires VE-cadherin and vascular endothelial growth factor receptors and activates an integrin-RhoA pathway (also different from this present manuscript). Coon et al (JCB, 2015) showed that VE-cadherin association with VEGFR2 and VEGFR3 is required to form the core mechanosensory complex in vitro and in vivo. A much more likely explanation for the different observations in epidermis is the nature of the mechanical input and the possible influence of other adhesion proteins such as P-cadherin and integrins that may have different effects in different tissue contexts (see Wang et al PNAS, 1998 p14821).

Related to the above question and to one of the other Reviewers' points, integrins were not considered in this study. What role are integrins playing? The authors addressed this by focusing on focal adhesions and paxillin staining, but integrins can be activated by intracellular signals as well as by secreted extracellular matrix via mechanisms that do not involve focal adhesion maturation. Rosario et al (2009) Dev Biol showed that secreted fibronectin affected cell polarity and cell movements in Xenopus embryos, and other reports implicated secreted laminin in integrin-dependent autocrine signaling.

Both integrins and P-cadherin were considered in our study. We have inactivated vinculin, which, in addition to being recruited to adherens junctions, is also an important mechanical linker at focal adhesions. However, loss of vinculin did not alter apical tension or tight junction formation, unlike loss of E-cadherin. Moreover, it is critical to emphasize that extracellular matrix is not present, and thus integrins do not engage in effective adhesion and are transcriptionally downregulated in the layer in which tight junctions are formed, and therefore this was not a meaningful aspect to further focus on.

As already mentioned in our first answer to the reviewer's point, where she/he indicated satisfaction to our previous round of revision on this issue, overexpression of P-cadherin in E-cadherin^{-/-} keratinocytes rescued tight junctions, indicating similar mechanical properties for these cadherins in the formation of tight junctions.

In the discussion we further acknowledge that E-cadherin may form a mechanosensitive complex with EGFR, similar to what has been observed for VE-cadherin and VEGFR, which also results in inhibition of VEGFR, similar to what we observe for EGFR. In contrast, in the study of Muhamed et al (2016), E-cadherin seems to not inhibit but to activate EGFR. One crucial difference in this study is that these experiments were done on single cells where cadherins were

only engaged through a ligand coated bead, therefore not mechanically coupled the actomyosin cytoskeleton of two neighboring cells, as is normally the case in adherens junctions.

The formation of mature desmosomes in their cultures and in vivo does raise the question of the role of keratin in the cortical tension measurements. If desmosomes require E-cadherin, then E-cadherin would indirectly regulate keratin filaments, which also likely contribute to cortical stiffness. They mention keratin in the response, but did not specifically address this.

We fully agree with the reviewer that the role of keratins and desmosomes in the regulation of tension and tight junction formation is an interesting one albeit beyond the scope of this paper. We have therefore added a small paragraph on the potential role of desmosomes and keratins on tension in epidermis to the discussion. Importantly, upon loss of E-cadherin desmosomes are formed, as previously stated, both in vivo and in vitro (see also Tunggal et al, 2005, Michels et al., JID, 2009). It is actually remarkable that there is a reduction in apical cortical tension as desmosomes are still present, which we now also state in the results section.

REVIEWERS' COMMENTS:

Reviewer #1 (Remarks to the Author):

Based on the authors' comments addressing the concerns raised, and on the efforts made to experimentally address the questions raised to the extent possible, I am satisfied with the changes.

Final answer to reviewer #1

REVIEWERS' COMMENTS:

Reviewer #1 (Remarks to the Author):

Based on the authors' comments addressing the concerns raised, and on the efforts made to experimentally address the questions raised to the extent possible, I am satisfied with the changes.

We are very happy to see that the reviewer is satisfied with textual and experimental changes to address his/her questions and comments.